# From Steatosis to Immunosenescence: The Impact of Metabolic Dysfunction on Immune Aging in HIV and Non-HIV Populations

**DOI:** 10.3390/biomedicines13102513

**Published:** 2025-10-15

**Authors:** Carlo Acierno, Maria Frontuto, Giulio Francesco De Stefano, Ana Erezanu, Andrea Limone, Simona Morella, Francesco Picaro, Donatella Palazzo, Michele Gilio

**Affiliations:** Department of Infectious Diseases, San Carlo Hospital, 85100 Potenza, Italy; mariolina.frontuto@ospedalesancarlo.it (M.F.); giulio.destefano@ospedalesancarlo.it (G.F.D.S.); ana.erezanu@ospedalesancarlo.it (A.E.); andrea.limone@ospedalesancarlo.it (A.L.); simona.morella@ospedalesancarlo.it (S.M.); francesco.picaro@ospedalesancarlo.it (F.P.); donatella.palazzo@ospedalesancarlo.it (D.P.); michele.gilio@ospedalesancarlo.it (M.G.)

**Keywords:** immunosenescence, HIV, MASLD, type 2 diabetes mellitus, metaflammation, epigenetic aging, gut–liver axis, TyG index, immune resilience, vaccine response

## Abstract

Immunosenescence, defined as the progressive decline of immune function with age, is increasingly recognized as a determinant of morbidity in people living with HIV (PLWH) and in individuals with metabolic dysfunction. The coexistence of chronic viral infection and systemic metabolic alterations—including metabolic dysfunction-associated steatotic liver disease (MASLD) and type 2 diabetes mellitus—creates a pro-inflammatory state (“metaflammation”) that accelerates immune aging. This narrative review synthesizes current clinical, translational, and experimental evidence on the cellular, molecular, and metabolic mechanisms underlying immunosenescence in HIV-positive and HIV-negative populations with metabolic dysfunction. Key converging pathways include chronic inflammation, mitochondrial dysfunction, microbial translocation, and altered immunometabolic signaling, leading to features such as CD8^+^CD28^−^ T-cell expansion, reduced CD4/CD8 ratios, and impaired vaccine responses. Biomarkers such as iAge, IMM-AGE, and the triglyceride–glucose (TyG) index have emerged as promising tools to quantify immune aging beyond chronological measures. Understanding these interconnected mechanisms offers opportunities for targeted interventions—such as metabolic reprogramming, microbiota modulation, and geroscience-based strategies—aimed at preserving immune resilience and promoting healthy aging in high-risk populations.

## 1. Introduction

Aging of the immune system represents one of the emerging challenges in contemporary medicine, particularly in contexts marked by complex comorbidities and multifactorial interactions. Within this framework, the intersection between metabolic dysfunction and immunosenescence is gaining increasing prominence, delineating a pathogenic axis capable of profoundly influencing the trajectory of biological aging. This narrative review aims to critically examine the cellular and molecular mechanisms underlying this interaction, with particular focus on people living with HIV (PLWH), in whom immune decline manifests in an earlier and more pronounced form.

The review is based on a rigorous selection of original clinical, translational, and molecular studies published over the past decade across seven key thematic domains (HIV, aging, immunosenescence, metabolic dysfunction, MASLD, metabolic syndrome, and hepatology). Contributions lacking empirical support (e.g., editorials, commentaries, opinion pieces) were excluded, and no quantitative meta-analysis was conducted. The objective is to provide an integrated and up-to-date synthesis of current evidence to better understand the etiopathogenesis of premature immune aging, particularly in high-risk clinical populations such as PLWH.

Immunosenescence is a gradual, multifactorial process that affects both innate and adaptive immunity, leading to phenotypic, functional, and epigenetic alterations that compromise the immune response to infectious, neoplastic, and vaccine-related stimuli [1]. Although traditionally associated with chronological aging, growing evidence supports the role of non-chronological accelerators—including metabolic dysfunction and HIV infection—via shared inflammatory and metabolic mechanisms [2,3].

Metabolic dysfunction—which encompasses insulin resistance, visceral obesity, hepatic steatosis (MASLD), and type 2 diabetes—has emerged as a principal driver of early-onset immunosenescence [4]. Visceral adipose tissue functions as an immunoactive organ, promoting chronic inflammation through the infiltration of senescent cells and the secretion of pro-inflammatory mediators (IL-6, TNF-α, leptin), with consequent inflammasome activation, reactive oxygen species (ROS) production, and mitochondrial damage [5].

In PLWH, immune aging is further exacerbated by infection-intrinsic factors, even under virologic suppression: a reduced CD4/CD8 ratio, accumulation of CD8^+^CD28^−^ T cells, depletion of the naïve T-cell compartment, chronic immune activation, and microbial translocation are hallmarks of an accelerated and dysfunctional immune phenotype [6]. Rather than acting independently, HIV infection and metabolic dysfunction appear to converge along a synergistic axis that accelerates immune decline and increases the risk of infections, malignancies, and multimorbidity [7].

The recent redefinition of hepatic steatosis as MASLD (metabolic dysfunction-associated steatotic liver disease) has reinforced the hypothesis of a systemic inflammatory continuum in which the liver acts not only as a target but also as an amplifier of immunometabolic dysfunction [8].

In PLWH with MASLD, the gut–liver–immune axis appears profoundly disrupted, characterized by dysbiosis, impaired mucosal barrier integrity, chronic hepatic inflammation, and marked immunosenescence [9,10].

In response to this scenario, novel tools have recently been proposed to quantify biological immune risk, such as “epigenetic clocks” (e.g., GrimAge, iAge) and predictive cellular phenotypes (e.g., IMM-AGE), which can overcome the limitations of chronological age and provide individualized estimates of immune aging [11]. Biomarkers such as CXCL9, IL-1β, and TNF-α correlate with clinical parameters of frailty, cardiovascular risk, and functional decline, particularly in individuals with HIV infection and metabolic dysfunction [12].

In light of these considerations, the present review aims to systematically and integratively explore the major pathophysiological determinants of immune aging, with particular reference to MASLD, type 2 diabetes, and metaflammation. Emerging markers of senescence, clinical implications in terms of infectious risk and vaccine responsiveness, and strategies for promoting immune resilience will also be discussed. The analysis, structured into eight sections, seeks to propose an interpretive paradigm for assessing and managing immunometabolic risk in PLWH and the general population.

The early convergence of HIV infection and metabolic dysfunction sets the stage for an accelerated trajectory of immune aging in PLWH. Key drivers—such as visceral adiposity, microbial translocation, and T-cell senescence—act through interconnected inflammatory and mitochondrial pathways to promote chronic immune activation and frailty (Table 1).

### Search Strategy

Given the narrative nature of this review, no formal systematic review protocol or PRISMA flow diagram was employed. Instead, we aimed to provide an updated and integrative synthesis of evidence on metabolic dysfunction and immune aging in both HIV and non-HIV populations. A comprehensive search was conducted in PubMed, Scopus, and Web of Science for articles published between January 2000 and July 2025, using combinations of the keywords “HIV”, “metabolic dysfunction”, “steatosis”, “MASLD”, “MAFLD”, “immunosenescence”, “microbiome”, “microRNA”, “gut permeability”, and “immune activation”. Reference lists from selected papers were also screened for additional relevant studies. Eligible works included original clinical, translational, and molecular studies; editorials, commentaries, and opinion pieces were excluded. The selection was based on the authors’ expertise and relevance to the review’s scope, without language restrictions.

## 2. Metabolic Dysfunction and Low-Grade Chronic Inflammation

### 2.1. MASLD/MAFLD: Epidemiology and Pathophysiology

Metabolic dysfunction-associated steatotic liver disease (MASLD), recently redefined to replace the former term “NAFLD”, represents the hepatic manifestation of a systemic, multifactorial metabolic dysfunction [13].

In 2020, an international panel of experts introduced the term metabolic dysfunction-associated fatty liver disease (MAFLD) to overcome the limitations of the NAFLD definition, which was based on exclusionary criteria (absence of significant alcohol consumption and other known causes of liver disease), instead favoring positive criteria linked to the presence of metabolic dysfunction [14].

More recently, in 2023–2024, international scientific societies further refined this terminology by replacing MAFLD with MASLD, with the aim of emphasizing the steatotic component and encompassing the diverse clinical presentations of metabolic dysfunction-related liver disease [15].

This terminological evolution reflects a conceptual shift: from merely excluding other etiologies (NAFLD) to actively identifying a metabolically at-risk phenotype (MAFLD/MASLD), with significant implications for risk stratification and clinical management [16].

The new MASLD definition, grounded in positive diagnostic criteria (visceral obesity, insulin resistance, dyslipidemia, hypertension), enables more accurate stratification of metabolic risk by integrating the hepatic phenotype into the broader context of metabolic syndrome [17].

Globally, the prevalence of MASLD ranges from 25% to 45%, exceeding 50% among individuals with type 2 diabetes mellitus [18].

Among PLWH, application of MAFLD criteria has revealed an estimated prevalence of 30% to 40%, even in patients receiving effective antiretroviral therapy with suppressed viremia, suggesting a pathogenic interplay between metabolic dysfunction and chronic inflammation [19].

The pathogenesis of MASLD is driven by the abnormal accumulation of triglycerides and free fatty acids (FFAs) in the liver, promoted by a positive energy balance, insulin resistance, and chronic visceral lipolysis [20].

Excessive hepatic FFA flux induces lipotoxicity, generates reactive oxygen species (ROS), impairs mitochondrial function, and activates inflammatory pathways through Toll-like receptors (TLRs) and the NLRP3 inflammasome [21].

This triggers a sterile inflammatory response that fuels fibrogenesis and progression toward metabolic steatohepatitis (MASH), cirrhosis, and hepatocellular carcinoma (HCC) [22,23].

Immunologically, MASLD is characterized by hyperactivation of innate immunity and loss of hepatic immune tolerance [24].

Damaged mitochondria release damage-associated molecular patterns (DAMPs) that activate Kupffer cells, resident macrophages, recruited monocytes, and dendritic cells, leading to the production of IL-1β, TNF-α, and IL-6 [25].

Chronic activation of hepatic stellate cells by these cytokines promotes collagen deposition and stiffening of the hepatic microenvironment [26].

Concurrently, the steatotic microenvironment becomes a site of infiltration by activated CD8^+^ T cells and NKT cells exhibiting an exhausted and senescent phenotype, with diminished cytotoxic function and upregulation of inhibitory markers (CD57, PD-1) [27].

In PLWH, this hepatic immune dysfunction intersects with HIV-induced systemic senescence, creating a complex clinical scenario in which MASLD acts both as an epiphenomenon and as a co-driver of the dysregulated immune response [28].

Mitochondrial dysfunction in MASLD, evidenced by reduced expression of key genes involved in mitochondrial biogenesis (PGC-1α, TFAM, MFN2) and altered dynamics of fission and fusion, constitutes a direct link to immune senescence [29].

These defects increase oxidative stress, activate apoptotic pathways, and impair the metabolic plasticity of immune cells [30].

Finally, in PLWH with MASLD, reductions in the CD4/CD8 ratio, expansion of senescent T lymphocytes, and heightened systemic inflammaging have been observed, which are conditions associated with diminished quality of life and increased multimorbidity [31].

These findings indicate that hepatic steatosis not only reflects global metabolic dysfunction but may also amplify immunosenescence trajectories in PLWH [32].

### 2.2. Type 2 Diabetes Mellitus: Insulin Resistance and Immune Response

Type 2 diabetes mellitus (T2DM), now recognized as one of the most prevalent chronic metabolic diseases worldwide, is closely linked to insulin resistance, adipocyte dysfunction, and chronic low-grade inflammation [33].

This pathogenic triad—comprising insulin resistance, adipocyte dysfunction, and chronic low-grade inflammation—not only contributes to the development of metabolic and vascular complications but also promotes premature immunosenescence [2].

Individuals with T2DM exhibit an accumulation of senescent T cells (CD28^−^/CD57^+^ or CCR7^−^/CD45RA^−^), characterized by reduced proliferative capacity, impaired migratory function, and heightened production of pro-inflammatory cytokines such as TNF-α [34].

This premature “inflammaging” phenotype of the immune system leads to functional deterioration of both adaptive and innate immunity, increasing susceptibility to infections, diminishing vaccine responsiveness, and disrupting the immunometabolic microenvironment [35].

In T2DM, the expansion of visceral adipose tissue constitutes a persistent source of inflammatory signals mediated by cytokines (TNF-α, IL-6), chemokines (MCP-1), and dysfunctional adipokines (resistin, RBP4), which interfere with insulin signaling and sustain a state of systemic metaflammation [36].

This inflammatory milieu exerts a profound influence on the composition and functionality of both innate and adaptive immune compartments [37].

At the cellular level, T2DM is marked by expansion of CD8^+^ effector memory T cells (TEMRA), progressive depletion of naïve CD4^+^ and CD8^+^ T cells, increased frequencies of senescent CD28^−^ T cells, and reductions in regulatory T- and B-cell populations—phenotypic hallmarks associated with immunosenescence [38,39].

These immune alterations are further exacerbated by mitochondrial bioenergetic dysfunction: individuals with T2DM exhibit impaired oxidative phosphorylation, increased ROS production, and dysregulation of mitophagy, biogenesis, and mitochondrial dynamics (fission/fusion), resulting in the accumulation of damaged mitochondria and release of DAMPs [40,41].

Such metabolic perturbations activate intracellular inflammatory pathways (JNK, NF-κB) and amplify NLRP3 inflammasome activation, promoting a dysfunctional, pro-senescent immune response [42,43].

The relationship between T2DM and immunosenescence is particularly significant in PLWH. In this population, the coexistence of insulin resistance, chronic inflammation, and immune dysregulation creates a deleterious synergy that accelerates the loss of TCR repertoire diversity, expansion of exhausted CD8^+^CD28^−^ T cells, and impairment of vaccine responses [44,45].

An emerging tool for assessing the impact of T2DM on immune aging in PLWH is the triglyceride–glucose (TyG) index, a surrogate marker of insulin resistance associated with reduced CD4/CD8 ratio, increased frequencies of senescent CD8^+^ T cells, and greater clinical frailty [32,46,47].

The combined pathogenesis of T2DM, hepatic dysfunction, and HIV establishes a high-risk immunometabolic axis characterized by the following:Chronic hepatometabolic inflammation, sustaining continuous release of pro-inflammatory cytokines (IL-6, TNF-α) and ROS, thereby perpetuating insulin resistance and metabolic impairment [48];Depletion and dysfunction of regulatory T cells (Tregs), driven by both HIV infection and hepatic inflammation, which limits the capacity to restrain immune activation and facilitates persistent sterile inflammation [49];Reduced immunological resilience, with progressive immunosenescence manifesting as phenotypic alterations (decreased CD4/CD8 ratio, expansion of exhausted T cells) and chronic activation biomarkers (sCD14, IL-6), intensified by the coexistence of metabolic and viral comorbidities [50].

These patients exhibit heightened vulnerability to infections, suboptimal vaccine responses, and age-related comorbidities, necessitating an integrated and personalized clinical approach [51,52,53].

Figure 1 summarizes the interplay between insulin resistance, altered gut microbiota, hepatic steatosis, and immune dysfunction in PLWH, highlighting the contribution of microbial translocation, mitochondrial stress, and CD8^+^CD28^−^ T-cell expansion to systemic immunosenescence.

### 2.3. Metaflammation and Chronic Systemic Inflammation

Metaflammation, defined as a chronic low-grade inflammatory state associated with nutritional excess and metabolic dysfunction, represents a common pathogenic denominator linking MASLD, type 2 diabetes mellitus (T2DM), and immunosenescence [54].

Unlike acute inflammation, which is targeted and self-limiting, metaflammation is characterized by a persistent, sterile, and systemic response, triggered by endogenous metabolic and mitochondrial signals rather than pathogenic stimuli [55].

Central mediators of metaflammation include damage-associated molecular patterns (DAMPs) originating from damaged mitochondria—such as oxidized mitochondrial DNA, extracellular ATP, reactive oxygen species (ROS), and oxidized lipids—as well as circulating free fatty acids [56].

These signals activate pattern-recognition receptors (PRRs), including TLR9, the NLRP3 inflammasome, purinergic receptors (P2X7), and RAGE on immune and endothelial cells, initiating a maladaptive, chronic sterile inflammatory response characteristic of metabolic-inflammatory disorders [57].

This inflammatory circuit drives activation of the JNK and NF-κB pathways, sustains production of IL-6, IL-1β, and TNF-α, and promotes recruitment of effector immune cells into metabolically active tissues, particularly the liver and visceral adipose tissue [58].

Immunologically, metaflammation induces progressive remodeling of the immune system: a decline in naïve cell populations, clonal expansion of terminally differentiated CD8^+^ T cells (CD28^−^), reduced TCR diversity, and diminished production of regulatory T and B cells are commonly observed [59].

These phenotypes are associated with impaired responses to antigenic stimuli and a heightened propensity for inflammatory cytokine secretion, constituting a profile of functional immune senescence [60,61].

In individuals with HIV, metaflammation is a key factor sustaining systemic inflammation despite virological suppression [62].

Damage to the intestinal barrier and microbial translocation contribute to chronic PRR activation, overlapping with mechanisms of metabolic dysfunction and amplifying the inflammatory response [63,64].

In this context, plasma markers such as sCD14, IL-6, and LBP correlate with frailty, cognitive decline, and non-AIDS-related comorbidities [65].

Metaflammation also contributes to vascular dysfunction, promoting arterial stiffness, endothelial senescence, and mitochondrial impairment, forming an axis that links chronic inflammation, vasculopathy, and immunosenescence [66,67].

This systemic state underlies the increased incidence of cardiovascular and cerebrovascular diseases observed in PLWH and in individuals with MASLD or T2DM [8,68,69].

At the molecular level, metaflammation drives a progressive loss of immune regenerative capacity, marked by hematopoietic stem cell exhaustion and disruption of the bone marrow niche—phenomena well documented in models of accelerated aging [70].

Moreover, chronic inflammasome activation in myeloid cells and inhibition of cellular autophagy amplify the accumulation of cytotoxic components and exacerbate senescence [71].

Ultimately, the convergence of metabolic dysfunction, HIV infection, and chronic systemic inflammation delineates a trajectory of accelerated and dysfunctional aging, marked by diminished immune resilience and increased clinical frailty [72].

This framework supports early intervention on metaflammation as a potential therapeutic lever to modulate immunological risk and improve outcomes in the most vulnerable populations.

These converging mechanisms underscore metaflammation as a central amplifier of systemic immunosenescence, linking metabolic dysfunction, chronic inflammation, and impaired immune resilience (Figure 2).

Table 2 summarizes the main immunometabolic alterations observed in PLWH with MASLD and T2DM, highlighting the key inflammatory mediators and their impact on immune aging. It also illustrates the role of the TyG index as an emerging biomarker for immunosenescence in this high-risk population (Table 2).

## 3. Immunosenescence: Cellular and Molecular Mechanisms

### 3.1. Senescence of the Innate and Adaptive Immune System

Immunosenescence is a profound and progressive transformation of the immune system, involving both innate and adaptive immunity, and results in a diminished ability to respond to pathogenic, neoplastic, and vaccine-derived stimuli [1].

Although physiologically associated with chronological aging, immunosenescence may manifest prematurely in the presence of metabolic dysfunction, HIV infection, or chronic systemic inflammation, delineating trajectories of accelerated aging [73].

Within the innate immune compartment, neutrophils exhibit progressive functional decline: reduced chemotaxis, attenuated phagocytic activity, impaired ROS production, and defective formation of neutrophil extracellular traps (NETs), leading to ineffective containment of bacterial and fungal infections [74].

Natural killer (NK) cells, despite increasing in number, undergo a decline in cytotoxicity and IFN-γ production, compromising surveillance against viral infections and tumors [75].

Dendritic cells, particularly plasmacytoid dendritic cells (pDCs), are both numerically reduced and functionally impaired, with diminished type I interferon production and a reduced ability to activate naïve T lymphocytes, contributing to viral vulnerability in older adults and PLWH [76].

Progressive thymic involution, characterized by replacement of lymphoid parenchyma with adipose tissue, constitutes a central driver of adaptive immunosenescence [77].

This leads to reduced production of naïve T cells, loss of T-cell receptor (TCR) diversity, and expansion of terminally differentiated memory T-cell clones, particularly CD8^+^CD28^−^ cells, expressing senescence markers such as KLRG1 and CD57, with high inflammatory cytokine output and resistance to apoptosis [78].

The B-cell compartment is also altered, with a reduction in the naïve subset, expansion of age-associated B cells (ABCs), and impaired ability to produce high-affinity antibodies [79].

ABCs, characterized by expression of T-bet and CD11c, exhibit ambiguous regulatory functions and are associated with autoimmune phenomena and poor vaccine responsiveness [80].

In PLWH, immunosenescence manifests more acutely and at an earlier age: chronic immune activation, premature depletion of naïve T cells, and persistent systemic inflammation drive the accumulation of CD8^+^CD28^−^CD57^+^KLRG1^+^ cells with an exhausted phenotype and dysfunctional effector capacity [81].

Even in the context of effective antiretroviral therapy (ART), these immunological alterations persist over time, suggesting that HIV acts more as an accelerator of the senescence process than as a merely reversible trigger [82].

These innate and adaptive immune dysfunctions converge in the loss of “integrated immunocompetence”, a concept describing the immune system’s capacity to mount a coordinated and adaptive response to external stimuli [83].

In older individuals and PLWH, fragmentation of this response is associated with increased susceptibility to infections, reduced vaccine efficacy, and a heightened risk of immune-inflammatory comorbidities [84].

In summary, immunosenescence entails a multilayered remodeling of the immune system, which in PLWH and individuals with metabolic dysfunction occurs prematurely and with more profound systemic impact [1].

Understanding these mechanisms is foundational for identifying early biomarkers and developing targeted therapeutic strategies aimed at enhancing immune resilience.

### 3.2. Molecular Markers and Cellular Phenotypes of Immune Aging

The identification of molecular and cellular markers of immunosenescence is a critical step toward evaluating the biological age of the immune system, predicting clinical vulnerability, and developing personalized interventions [85].

These markers reflect stable, measurable phenotypic changes in lymphocyte subsets, epigenetic and transcriptional patterns, and circulating inflammatory signals that accumulate over time and under conditions of chronic stress, such as HIV infection or metabolic dysfunction [86].

In the T-cell compartment, senescence is marked by loss of the co-stimulatory molecule CD28, along with the emergence of inhibitory receptors such as KLRG1, CD57, and NKG2D on terminally differentiated CD8^+^ lymphocytes [87].

These cells exhibit low replicative potential, overproduction of IFN-γ and TNF-α, heightened resistance to apoptosis, and a persistent pro-inflammatory phenotype [88].

In PLWH, these populations are expanded even in individuals with suppressed viremia, indicating a lasting disruption in the balance between effector and regulatory immune responses [89].

In the B-cell compartment, age-associated B cells (ABCs) represent an emerging subpopulation linked to aging [90].

Characterized by expression of T-bet, CD11c, and CD95, ABCs display reduced capacity to differentiate into antibody-producing plasma cells and an increased tendency to secrete autoreactive IgM, contributing to autoimmunity and suboptimal vaccine responses [91].

At the epigenetic level, the use of “molecular clocks” has enabled estimation of biological age via DNA methylation at specific CpG sites [92].

The main validated epigenetic clocks include Horvath, Hannum, and GrimAge, the latter being particularly predictive of cardiovascular morbidity and mortality [11,93].

In PLWH, significant epigenetic acceleration has been observed, correlating with infection duration, CD4^+^ nadir, and persistent systemic inflammation [94].

Concurrently, the integration of immunophenotypic and transcriptomic data has led to the development of composite metrics such as the IMM-AGE score, which reflects the functional state of the immune system and correlates more strongly than chronological age with frailty and multimorbidity [88].

The iAge index, in particular, integrates key inflammatory cytokines (CXCL9, IL-6, IL-1β), tissue damage biomarkers, and transcriptomic data and has been proposed as a predictive parameter of accelerated immune aging in PLWH with metabolic dysfunction [95].

Other relevant markers include telomere length in CD4^+^ and CD8^+^ lymphocytes [96]; mitochondrial senescence (e.g., reduced expression of PGC-1α, TFAM) [97]; and plasma concentrations of pro-inflammatory chemokines (CXCL10, CCL2) and endothelial mediators (sVCAM-1, ICAM-1) [98], all associated with increased incidence of cardiovascular events and immunosenescence [99].

Importantly, in individuals with T2DM or MASLD, several markers of immune aging are already altered even in the absence of HIV infection [100].

For instance, the triglyceride–glucose (TyG) index shows a negative correlation with the CD4/CD8 ratio and a positive correlation with the expansion of CD8^+^CD57^+^KLRG1^+^ T cells, providing a measurable link between metabolic dysfunction and systemic immunosenescence [32].

These markers offer valuable tools for monitoring clinical risk, guiding personalized interventions, and evaluating the effectiveness of geroscience-based strategies in individuals with immunometabolic vulnerability [101].

These features of immune aging are synthesized in Table 3, which summarizes the main senescent phenotypes, molecular markers, and functional consequences observed in both innate and adaptive compartments, with particular emphasis on their amplification in people living with HIV (PLWH). This integrative overview supports the relevance of using composite metrics and phenotypic markers to assess immune aging beyond chronological parameters.

### 3.3. MicroRNAs as Emerging Biomarkers

Alongside advances in microbiome research, analysis of microRNA (miRNA) expression profiles is emerging as a promising and innovative approach to understanding and monitoring immunometabolic alterations in both HIV-positive individuals and non-HIV subjects with metabolic dysfunction [102].

MicroRNAs are short (19–24 nucleotides), highly conserved, endogenous non-coding RNA sequences that regulate gene expression at the post-transcriptional level by binding to the 3′-UTR of target mRNAs, leading to degradation or translational inhibition [103]. They play a critical role in regulating multiple pathogenic processes implicated in immunosenescence and metabolic complications.

In particular, miRNAs modulate chronic inflammation by controlling the production of pro- and anti-inflammatory cytokines and by interacting with key intracellular pathways, including NF-κB, JAK/STAT, and Toll-like receptors (TLRs) [104,105].

They also influence lipid and glucose metabolism by regulating genes involved in lipogenesis and glycolysis, as well as glucose transporters, as exemplified by miR-33 and miR-143 [106].

Another area of interest is hepatic fibrosis, where miRNAs modulate hepatic stellate cell activation and extracellular matrix deposition through mechanisms involving the TGF-β/Smad signaling pathway [107,108].

Emerging evidence also suggests a role for miRNAs in regulating adaptive immune responses, including lymphocyte differentiation and plasticity, and activation of monocytes and macrophages [109,110].

In the context of chronic inflammation and dysbiosis associated with digestive and metabolic diseases, specific miRNAs show marked dysregulation and acquire key pathogenic roles. MiR-122, for example, has emerged as an early hepatic biomarker, being highly expressed in the liver and associated with both steatosis and fibrosis, thus signaling early-stage liver dysfunction [111,112]. MiR-21 and miR-192, like miR-122, are associated with insulin resistance and hepatic fat accumulation [108].

Simultaneously, miR-155, miR-146a, and miR-223 modulate intestinal inflammatory responses and maintain mucosal barrier integrity: miR-146a restricts NF-κB pathway activation, while miR-223 regulates granulocyte function and IL-1β production [104,105].

In the transition toward neoplastic processes, miR-31 contributes to epithelial regeneration via the Wnt pathway, and miR-375 modulates mucosal proliferation and pancreatic function, suggesting a potential link between chronic inflammation and oncogenesis [113].

Due to their remarkable stability in biological fluids—such as serum, plasma, saliva, urine, and feces—and their detectability through sensitive, non-invasive molecular techniques (e.g., RT-qPCR, next-generation sequencing, and microarray analysis), microRNAs are well positioned as ideal diagnostic and prognostic biomarkers [114,115].

Their expression profiles can distinguish disease stages and phenotypes with high specificity—for instance, discriminating between Crohn’s disease and ulcerative colitis [116] or identifying subgroups of patients with MASLD (metabolic dysfunction-associated steatotic liver disease) who exhibit a predominant inflammatory phenotype [111].

Beyond differential diagnosis, miRNAs are proving valuable in predicting disease progression, such as the transition from simple steatosis to nonalcoholic steatohepatitis (NASH) or early detection of advanced hepatic fibrosis [111].

Longitudinal measurement of circulating miRNAs also offers a promising strategy for dynamically monitoring responses to gut microbiota-targeted interventions, including nutraceuticals, probiotics, prebiotics, and conventional pharmacological therapies.

Another compelling area of interest is the bidirectional relationship between the microbiome and host miRNAs: dysbiosis can alter host miRNA expression, thereby affecting metabolic and immune pathways, while host-derived miRNAs can in turn modulate the composition of the gut microbiota [117].

Integration of miRNA and microbiome profiles may thus yield a more comprehensive and predictive framework for assessing the risk of immunosenescence and metabolic complications.

This perspective paves the way for precision medicine strategies, in which the combined analysis of molecular and microbial biomarkers can guide patient stratification and inform personalized therapeutic interventions.

### 3.4. Inflammaging: Origins, Systemic Impact, and Predisposing Factors

The term inflammaging describes a state of chronic, systemic, low-grade sterile inflammation that physiologically accompanies aging but which can be exacerbated by exogenous and pathological factors such as metabolic dysfunction and HIV infection [118].

This persistent inflammatory state represents not only a molecular hallmark of immunosenescence but also an active driver of tissue damage, immune dysfunction, and increased susceptibility to age-related chronic diseases [119].

The genesis of inflammaging is multifactorial. At the molecular level, it arises from the chronic secretion of pro-inflammatory cytokines (IL-6, TNF-α, IL-1β) by senescent immune cells exhibiting a senescence-associated secretory phenotype (SASP), as well as by parenchymal cells such as adipocytes and steatotic hepatocytes [120,121].

Cellular damage signals (DAMPs)—including oxidized mitochondrial DNA, extracellular ATP, and HMGB1—activate pattern-recognition receptors (PRRs) such as Toll-like receptors, the NLRP3 inflammasome, and the cGAS-STING pathway, perpetuating the inflammatory loop [122].

Inflammaging has profound systemic consequences. At the vascular level, it promotes endothelial senescence, microcirculatory dysfunction, and arterial stiffness, thereby facilitating cardiovascular events [123,124].

It also drives hematopoietic reprogramming: hematopoietic stem cells become skewed toward myeloid lineages at the expense of lymphopoiesis, due to inflammatory remodeling of the bone marrow. At the tissue level, chronic inflammation and SASP promote fibrotic deposition, selective apoptosis (favoring the survival of senescent cells), and infiltration by dysfunctional, pro-inflammatory effector cells, further impairing repair mechanisms and homeostasis [125].

In PLWH, inflammaging is amplified by viral and paraviral factors: microbial translocation due to mucosal barrier breakdown; cytomegalovirus (CMV) latency and reactivation, which promotes immune senescence; persistence of viral reservoirs with continuous antigen production; and chronic immune activation, evidenced by inflammatory markers such as IP-10, sCD14, sCD163, TNF-α, and IL-6 [126,127,128].

These interconnected processes sustain prolonged systemic inflammation that contributes to tissue damage and age-related comorbidities.

Even in the presence of suppressed viremia, inflammatory markers such as sCD14, IL-6, and D-dimer remain elevated and are predictive of adverse clinical events, frailty, and mortality [129,130].

Metabolic dysfunction acts as an additional predisposing factor.

Hepatic steatosis (MASLD) is associated with enhanced NLRP3 inflammasome activation, mitochondrial dysfunction, and secretion of pro-inflammatory chemokines by hepatocytes and infiltrating immune cells [131].

Type 2 diabetes mellitus further amplifies inflammaging via chronic hyperglycemia, oxidative stress, and gut dysbiosis, which promote activation of the innate immune system and loss of peripheral tolerance [10,132].

Genetic (e.g., polymorphisms in *IL6*, *TNFA*, *TLR4*), epigenetic (e.g., hypermethylation of anti-inflammatory genes, acceleration of epigenetic clocks), and environmental factors (e.g., physical inactivity, hypercaloric diet, smoking) also contribute to individual vulnerability [133,134].

In PLWH with an altered metabolic phenotype, this convergence drives an accelerated immune aging process that deviates from the physiological trajectory and shifts toward pathological senescence, characterized by the early onset of disability and multimorbidity [135].

In summary, inflammaging results from a synergistic convergence of metabolic, immune, viral, and environmental factors, progressively impairing immune plasticity and increasing susceptibility to exaggerated inflammatory responses. Early identification and modulation of modifiable factors—such as mitochondrial dysfunction and intestinal dysbiosis—represent strategic opportunities for preventive interventions capable of halting the progression toward frailty and pathological immunosenescence.

## 4. Immunometabolism: The Link Between Metabolism and Immune Senescence

### 4.1. Cellular Metabolism of Immune Cells Under Physiological Conditions

Cellular metabolism in immune cells constitutes a central regulatory mechanism of immune responses.

Through pathways such as aerobic glycolysis, oxidative phosphorylation (OXPHOS), fatty acid oxidation (FAO), and amino acid metabolism, immune cells dynamically modulate their functional, proliferative, and differentiation states [136].

Indeed, the balance between anabolic and catabolic metabolic programs determines the acquisition of effector, regulatory, memory, or inflammatory phenotypes in both lymphoid and myeloid subsets, thereby shaping immune responses under physiological and pathological conditions (e.g., infections, tumors, autoimmune diseases) [137].

Under physiological conditions, naïve immune cells—both T and B—maintain a quiescent metabolic state, primarily supported by mitochondrial oxidative phosphorylation and the tricarboxylic acid (TCA) cycle, which provide high energy efficiency with minimal reactive oxygen species (ROS) production [138].

Upon antigenic or cytokine-mediated activation, CD4^+^ and CD8^+^ T lymphocytes undergo a rapid metabolic switch from oxidative metabolism to aerobic glycolysis, even in normoxic conditions [139].

This metabolic reprogramming, which also involves activation of the pentose phosphate pathway (PPP), is essential for the generation of NADPH and ribose-5-phosphate—precursors required for nucleotide, lipid, and protein synthesis during proliferation and effector expansion [137].

This process is orchestrated by a signaling network that includes PI3K-AKT, mTORC1, HIF-1α, and c-Myc, which integrate extracellular stimuli with transcriptional regulation of metabolic and immunologic gene expression [140,141].

Metabolic polarization in T lymphocytes is tightly linked to their functional specialization: pro-inflammatory subsets (Th1, Th17) rely predominantly on aerobic glycolysis to sustain rapid expansion and anabolic biosynthesis, whereas regulatory T cells (Tregs) and memory T cells utilize more energy-efficient pathways based on fatty acid oxidation and mitochondrial OXPHOS—metabolic programs optimized for homeostatic maintenance and long-term persistence [142].

A similar dichotomy is observed in macrophages: the pro-inflammatory M1 phenotype is supported by glycolysis and intermediates such as succinate and citrate, while the anti-inflammatory M2 phenotype depends on OXPHOS and FAO [143].

The immunometabolic distinction between physiological and dysfunctional states is illustrated in Figure 3, highlighting how shifts in mitochondrial function, ROS production, and glycolytic dependence contribute to loss of immune plasticity and the emergence of senescent phenotypes.

Metabolic balance is not merely reflective of activation status; it is a causal determinant of immune function. Nutrient availability, oxygen tension, redox state, and mitochondrial charge directly influence cell survival, cytokine production, immunologic memory durability, and peripheral tolerance [137,144].

In healthy individuals, this metabolic architecture ensures an effective yet self-regulating adaptive immune response, capable of resolving inflammation without depleting cellular energy reserves [118,125].

However, under chronic stress conditions, such as HIV infection or metabolic dysfunction, this plasticity becomes impaired, leading to premature immune exhaustion, loss of functional memory, and the emergence of senescent phenotypes [127].

Immune physiology relies on finely tuned metabolic regulation, wherein the bioenergetics of resting immune cells—through a balanced use of glycolysis, OXPHOS, and FAO—are critical to adaptive immune efficiency. Under normal metabolic conditions, this equilibrium enables inflammation resolution through sustainable energy usage.

By contrast, in chronic metabolic disorders such as MASLD (metabolic dysfunction-associated steatotic liver disease) and T2DM, the altered metabolic environment—including nutrient imbalances, mitochondrial dysfunction, oxidative stress, and disrupted immunoregulatory pathways—results in the following:Reduced immune metabolic plasticity, with limited flexibility to shift between glycolysis and OXPHOS;Increased immunosenescence, with accumulation of senescent T lymphocytes (CD28^−^ CD57^+^) associated with chronic inflammation and loss of immune memory;Dysregulation of peripheral tolerance, resulting in a systemic pro-inflammatory state, further exacerbated by intestinal dysbiosis and visceral adiposity.

### 4.2. Metabolic Dysfunctions of Immune Cells in MASLD and Type 2 Diabetes Mellitus

In pathological conditions such as MASLD (metabolic dysfunction-associated steatotic liver disease) and type 2 diabetes mellitus (T2DM), the metabolic regulation of immune cells undergoes profound disruptions, shifting them toward pro-inflammatory, senescent, and metabolically exhausted phenotypes [145].

These dysfunctions emerge through converging signals—including insulin resistance, lipotoxicity, oxidative stress, and mitochondrial bioenergetic impairment—with significant systemic implications for the trajectory of immune aging [146].

In MASLD, hepatic accumulation of free fatty acids (FFAs) and triglycerides promotes a state of chronic lipotoxicity that deeply alters local immune metabolism [147].

Liver-resident immune cells, particularly CD8^+^ T lymphocytes and NKT cells, exhibit persistent activation, reduced cytotoxic function, increased expression of PD-1 and CD57, and loss of antigenic memory [148].

The steatotic hepatic microenvironment, enriched in ROS and pro-inflammatory mediators (IL-1β, IL-6), activates the NLRP3 inflammasome in both immune and parenchymal cells, amplifying the inflammatory response and promoting fibrogenesis [131].

Concurrently, mitochondrial dysfunction—evidenced by reduced expression of PGC-1α, TFAM, and MFN2—compromises the ability of T cells to sustain high-energy responses, resulting in metabolic exhaustion, impaired ATP production, and increased release of mitochondrial DAMPs [30].

This context drives polarization toward senescent immune phenotypes and loss of metabolic flexibility.

In T2DM, chronic hyperglycemia and systemic insulin resistance profoundly reshape immune metabolism. Effector T cells display a highly glycolytic yet inefficient profile, characterized by the persistent activation of mTOR and HIF-1α pathways and emergence of a SASP-like phenotype [149].

Chronic activation of intracellular inflammatory pathways, combined with reduced availability of efficient oxidative substrates, establishes a dysfunctional state of immunometabolic inflammation, further aggravated by depletion of regulatory T cells and reduced TCR repertoire diversity [150].

In individuals with coexisting MASLD and T2DM—a highly prevalent condition among PLWH—these alterations are synergistic, further impairing immune function. Increased expression of senescence markers (CD28^−^, CD57^+^) has been observed in T lymphocytes, associated with a reduced CD4/CD8 ratio and diminished quality of life [32].

Within this context, the triglyceride–glucose (TyG) index has emerged as a composite marker of immunometabolic risk, reflecting the interplay between insulin resistance, hepatic steatosis, and immune aging [29].

In summary, the metabolic dysfunction characterizing MASLD and T2DM profoundly undermines the bioenergetic foundation of immune cells, reducing their activation, persistence, and adaptive capacity. Prolonged exposure to a pro-inflammatory, metabolically misaligned microenvironment drives the progressive loss of immunologic plasticity, fostering the accumulation of senescent and exhausted phenotypes.

These alterations contribute to the establishment of a chronic state of inflammaging, with systemic repercussions on immune function. This underscores the urgent need for therapeutic strategies that go beyond glycemic or lipid control and target the metabolic reprogramming of immune cells. Such approaches, by modulating specific metabolic checkpoints (e.g., mTOR, AMPK, PGC-1α), may restore effector and regulatory functions, reactivate immunologic memory, and decelerate premature immune senescence.

A synthesis of these immunometabolic pathways is presented in Table 4, highlighting how metabolic stressors in MASLD and T2DM disrupt immune cell bioenergetics, favor senescence, and amplify systemic inflammation in PLWH.

This visual summary underscores the relevance of metabolic reprogramming as a therapeutic target in immunosenescence.

This immunometabolic trajectory is further summarized in the conceptual model shown in Figure 4 that integrates the progression from metabolic dysfunction to immune senescence, highlighting the roles of visceral adiposity, chronic inflammation, and mitochondrial impairment in shaping an exhausted immune phenotype.

### 4.3. Role of Adipocytes, Cytokines, and the Microbiota–Gut–Liver Axis

Adipose tissue is no longer viewed as a passive energy reservoir but as an active immunoendocrine organ capable of orchestrating systemic responses through the secretion of hormones, cytokines, and chemokines [151].

Under conditions of caloric excess, adipocyte hypertrophy, and visceral obesity—hallmarks of metabolic dysfunction and highly prevalent among PLWH—adipose tissue undergoes a profound pro-inflammatory transformation, with significant repercussions for immune homeostasis and the aging of the immune system [152].

Dysfunctional adipocytes secrete pro-inflammatory adipokines such as leptin and resistin, alongside reduced levels of adiponectin. These alterations directly affect T-cell activation, macrophage polarization, and dendritic cell function [153].

The resulting inflammatory microenvironment promotes the recruitment of M1 macrophages and the local production of TNF-α, IL-1β, and IL-6, activating intracellular inflammatory pathways such as NF-κB and the NLRP3 inflammasome [154].

In individuals with HIV, these processes are exacerbated by ART-induced lipodystrophy and lipid metabolic dysregulation, contributing to an accelerated state of metabolic immunosenescence [155].

Simultaneously, the gut–liver–immune axis plays a central role in propagating metaflammation [10].

In intestinal dysbiosis—a common feature of high-energy-density diets and various chronic conditions—there is reduced bacterial diversity, depletion of butyrate-producing commensals (e.g., *Faecalibacterium prausnitzii*, *Roseburias* spp.), and enrichment in pro-inflammatory species such as *Proteobacteria* and *Enterobacteriaceae* [10,156].

This dysbiosis disrupts intestinal barrier function, increasing epithelial permeability and systemic translocation of lipopolysaccharide (LPS) and microbial fragments (PAMPs), which reach the liver via the portal vein and activate PRRs on hepatocytes, Kupffer cells, and dendritic cells [157]. This process sustains chronic hepatic inflammation and stimulates the production of IL-1β, IL-6, and TGF-β, promoting fibrogenesis and contributing to both local and systemic immunosenescence [158].

Microbial metabolites—particularly short-chain fatty acids (SCFAs) such as acetate, propionate, and butyrate—directly modulate immune function through activation of GPR41/43 receptors expressed on T cells, dendritic cells, and adipocytes [159].

In PLWH, the depletion of these metabolites and impaired interaction with the intestinal immune system contribute to the loss of mucosal tolerance, expansion of senescent T cells, and systemic propagation of inflammaging [160].

Ultimately, in contexts characterized by adipocyte dysfunction, systemic cytokine release, and intestinal dysbiosis, the mechanisms of chronic inflammation and immunosenescence reinforce one another. The adipose–gut–liver axis emerges as a critical regulatory node in immunometabolism and the metabolic comorbidities observed in individuals with HIV.

Accordingly, combined therapeutic approaches—including nutritional interventions targeting the microbiota, anti-inflammatory pharmacologic strategies, and the use of prebiotics and probiotics—hold promise for mitigating immunosenescence and improving the immunometabolic profile of PLWH with metabolic dysfunction.

The interplay between gut dysbiosis, hepatic inflammation, and immune senescence in PLWH is illustrated in Figure 5, emphasizing the sequential disruption of mucosal and hepatic barriers and the emergence of senescent immune phenotypes.

These interactions within the gut–liver–immune axis are summarized in Table 5, which highlights the immunological consequences of adipose tissue dysfunction, dysbiosis, microbial translocation, and ART-related metabolic effects in PLWH. This comprehensive framework supports the rationale for targeting the gut–liver interface to mitigate systemic immunosenescence.

These immunological disruptions within the gut–liver–adipose axis are schematically illustrated in Figure 6, highlighting the bidirectional relationships driving systemic inflammaging in PLWH.

### 4.4. Gut Permeability and Systemic Inflammation in HIV and Non-HIV Individuals

Increased intestinal permeability (“leaky gut”) is a well-recognized and extensively studied contributor to systemic immune activation in both HIV-infected and HIV-negative individuals with metabolic dysfunction [10,161].

Under physiological conditions, the intestinal epithelial barrier—composed of a single layer of enterocytes sealed by tight junction proteins (e.g., claudins, occludin, and zonula occludens-1)—acts as the primary physical and immunological interface between the host and the gut microbiota [162].

In the context of chronic metabolic stress, inflammation, or viral infection, these tight junctions become structurally and functionally compromised, permitting translocation of microbial products into the lamina propria and systemic circulation [163].

Among these translocated products, lipopolysaccharides (LPS) from Gram-negative bacteria are particularly potent activators of innate immunity. LPS binds to LPS-binding protein (LBP) and CD14, triggering Toll-like receptor 4 (TLR4)-mediated signaling in monocytes, macrophages, and dendritic cells [164].

This activation leads to NF-κB pathway engagement, transcription of pro-inflammatory cytokines (TNF-α, IL-1β, IL-6), and upregulation of adhesion molecules and chemokines, thereby establishing a state of chronic low-grade inflammation or “metaflammation” [165].

Evidence from studies on metabolic syndrome shows that gut dysbiosis—characterized by reduced microbial diversity, depletion of beneficial commensals such as *Akkermansia muciniphila* and *Faecalibacterium prausnitzii*, and enrichment of pro-inflammatory taxa such as *Proteobacteria*, *Prevotella*, and *Klebsiella*—is closely linked to increased intestinal permeability [10].

These microbiota alterations are functionally associated with increased LPS release, enhanced NF-κB activation, NLRP3 inflammasome activation, and chronic stimulation of both myeloid and lymphoid immune compartments [164,165].

Functionally, these changes contribute to insulin resistance, endothelial dysfunction, mitochondrial stress, and progressive immune senescence [166].

In people living with HIV (PLWH), disruption of the gut barrier is an early event, detectable within days to weeks following primary infection, due to the massive depletion of Th17 CD4^+^ T cells in the gastrointestinal mucosa [167].

Even after prolonged antiretroviral therapy (ART), only partial restoration of mucosal immunity occurs, and microbial translocation persists despite full virologic suppression [167].

Circulating biomarkers such as soluble CD14 (sCD14), LBP, and intestinal fatty acid-binding protein (I-FABP) remain chronically elevated, reflecting ongoing antigenic exposure. This persistent stimulus perpetuates T-cell activation, reduces T-cell receptor (TCR) repertoire diversity, and maintains pro-inflammatory monocyte/macrophage phenotypes, thereby accelerating immunosenescence and increasing vulnerability to both infectious and non-infectious comorbidities [168].

Therapeutically, modulation of the gut microbiota and intestinal barrier integrity has emerged as a promising strategy to reduce systemic inflammation and restore immune homeostasis [169].

Nutritional interventions—such as high-fiber diets, polyphenols, and omega-3 fatty acids—along with prebiotics, probiotics, and synbiotics, have demonstrated the potential to increase the abundance of short-chain fatty acid (SCFA)-producing bacteria, strengthen epithelial integrity, and reduce pro-inflammatory cytokine expression [169].

In metabolic syndrome, fecal microbiota transplantation (FMT) has been shown to enhance insulin sensitivity, improve lipid profiles, and lower blood pressure, while promoting colonization by anti-inflammatory microbial taxa [63,170].

However, translating these approaches to PLWH requires careful consideration of their unique immunologic milieu, the impact of ART on microbiota composition, and the potential risks in immunocompromised hosts.

This interplay between gut permeability, dysbiosis, and systemic inflammation represents a critical and potentially modifiable driver of immunometabolic dysfunction, with significant implications for both HIV and non-HIV populations.

## 5. HIV and Accelerated Immune Aging

### 5.1. Chronic Immune Activation and Immunological Dysregulation in PLWH

HIV infection is characterized, from its earliest stages, by massive systemic immune activation, which persists even after the establishment of effective virologic suppression through antiretroviral therapy (ART) [126].

This state of chronic activation is one of the principal drivers of accelerated immune aging in people living with HIV (PLWH), independent of chronological age and absolute CD4^+^ T-cell count [171].

At the cellular level, HIV induces profound early depletion of activated CD4^+^ T cells, disrupting T-cell homeostasis and reducing T-cell receptor (TCR) repertoire diversity. Simultaneously, the CD8^+^ compartment undergoes marked hyperactivation, with expansion of senescent and terminally differentiated subpopulations (CD28^−^CD57^+^) characterized by exhausted phenotypes, high production of inflammatory cytokines (IFN-γ, TNF-α), and diminished proliferative capacity [172].

T-cell activation is further marked by co-expression of CD38 and HLA-DR, which remain elevated even during long-term ART and serve as independent indicators of clinical progression and immunosenescence [173].

Immune dysregulation in PLWH extends beyond the T-cell compartment. B-cell function is impaired, with reduced production of antigen-specific antibodies, loss of memory B cells, and accumulation of atypical and senescent subsets (ABCs), often associated with suboptimal vaccine responses [174].

Natural killer (NK) cells, although preserved in number, exhibit pronounced functional impairment, including reduced cytotoxicity, decreased expression of activating receptors (NKG2D, NKp30), and altered IFN-γ secretion, compromising antiviral and antitumor surveillance [175].

Dendritic cells (DCs), essential for the initiation of adaptive responses, are also reduced in number and functional efficiency, with impaired type I interferon production and antigen presentation capacity, further contributing to immunosenescence in PLWH [176].

The persistence of chronic immune activation in PLWH is exacerbated by latent viral coinfections, particularly cytomegalovirus (CMV), which drives clonal expansion of highly differentiated, senescent CD8^+^ T cells [177].

Ongoing immune activation is associated with increased incidence of non-AIDS-related conditions, including cardiovascular disease, malignancies, and frailty, even in young individuals with good numerical immune recovery [178].

Thus, HIV infection induces, from its early phases, a profound restructuring of the immune system, marked by persistent chronic activation, loss of lymphocyte repertoire diversity, and accumulation of senescent phenotypes. This dysfunctional immunological paradigm persists despite effective ART and is closely linked to non-AIDS comorbidities and accelerated immunosenescence in PLWH.

Accordingly, targeting residual immune activation emerges as a strategic therapeutic objective.

### 5.2. Microbial Translocation, Intestinal Barrier, and Persistent Inflammation

One of the most relevant mechanisms driving chronic immune activation and immunosenescence in PLWH is the early and persistent loss of intestinal barrier integrity.

As early as the initial stages of HIV infection, there is a rapid and selective depletion of mucosal CD4^+^ T cells, particularly Th17 cells, in the gastrointestinal tract, which irreversibly impairs barrier function and microbial homeostasis regulation [179].

Loss of epithelial integrity facilitates the systemic passage of microbial components—such as lipopolysaccharide (LPS), bacterial DNA, and peptidoglycans—which activate pattern recognition receptors (PRRs) on monocytes, macrophages, and dendritic cells through TLR4, TLR9, and NOD-like receptors, thereby stimulating the release of inflammatory cytokines (IL-6, TNF-α, IL-1β) [180].

This mechanism sustains a chronic state of systemic inflammation even in the absence of active viral replication, creating a favorable milieu for premature immunosenescence.

Plasma biomarkers of microbial translocation, including LPS-binding protein (LBP), intestinal fatty acid-binding protein (I-FABP), and soluble CD14 (sCD14), remain persistently elevated in ART-treated PLWH and are associated with immune activation, cardiovascular risk, frailty, and early mortality [181].

In parallel, the gut microbiota of PLWH displays a markedly dysbiotic profile: reduced bacterial diversity, depletion of anti-inflammatory butyrate-producing commensals (e.g., *Faecalibacterium prausnitzii*, *Akkermansia muciniphila*), and increased abundance of pro-inflammatory taxa (e.g., *Prevotella*, *Proteobacteria*) [182].

These changes contribute to ongoing PRR activation and chemokine production, perpetuating the vicious cycle of dysbiosis, microbial translocation, and systemic inflammation.

In PLWH, this inflammatory cascade leads to a profound restructuring of the immune compartment, characterized by expansion of senescent CD8^+^ T cells, elevated production of IFN-γ and TNF-α, and progressive loss of TCR repertoire diversity [183].

These alterations persist even after years of ART, suggesting that intestinal mucosal damage is not readily reversible [184].

This state of sterile chronic inflammation, amplified by an altered microbiome, contributes to increased risk of non-AIDS-related conditions including atherosclerosis, hepatic steatosis, and cognitive impairment which are all associated with systemic immunosenescence [185].

In conclusion, microbial translocation and dysbiosis are central drivers of immunosenescence in PLWH, acting independently of active viremia. Interventions aimed at restoring intestinal integrity and modulating microbiota could offer novel therapeutic avenues for slowing immune decline and enhancing systemic resilience in this population.

Table 6 summarizes the key mechanisms of chronic immune activation in PLWH, including persistent T-cell activation, dysregulation of B and NK cells, dendritic cell impairment, and microbial translocation, all contributing to accelerated immunosenescence and clinical vulnerability. This synthesis reinforces the rationale for immunomodulatory strategies beyond virologic suppression.

### 5.3. Interactions Between Metabolic Dysfunction and Immunosenescence in HIV-Positive Individuals

In PLWH, metabolic dysfunction acts as a potent amplifier of immunosenescence through a complex and bidirectional interaction with chronic immune activation induced by viral infection. The altered metabolic phenotype—already highly prevalent in this population—is the result of multiple factors: antiretroviral therapy-induced alterations (particularly protease inhibitors and NRTIs), persistent systemic inflammation, adipocyte dysfunction, and HIV-related epigenetic modifications [186].

In the presence of MASLD (metabolic dysfunction-associated steatotic liver disease) or type 2 diabetes, immune cells in PLWH display phenotypic features consistent with advanced senescence: CD8^+^CD28^−^CD57^+^ T cells, inefficient glycolytic polarization, hyperproduction of IFN-γ and TNF-α, and impaired ability to generate functional memory [187]. Dysfunctional adipose tissue—frequently observed in PLWH with lipodystrophy or visceral obesity—constitutes a chronic source of pro-inflammatory cytokines (IL-6, resistin) that activate the inflammasome and propagate metaflammation, with systemic effects on immune functionality [188].

This condition is associated with increased expression of mitochondrial metabolic exhaustion markers (e.g., reduced PGC-1α, elevated ROS), contributing to loss of metabolic plasticity and effector function in T cells [189].

Recent studies have shown that the triglyceride–glucose (TyG) index—a surrogate marker of insulin resistance—is significantly elevated in PLWH with MASLD or T2DM and correlates negatively with the CD4/CD8 ratio and positively with the frequency of senescent CD8^+^ T cells, suggesting a direct association between insulin resistance and immunosenescence [32]. This index has emerged as a potential integrated biomarker of immunometabolic risk in PLWH.

At the hepatic level, the coexistence of MASLD and HIV is associated with persistent infiltration of dysfunctional cytotoxic CD8^+^ T cells and activated monocytes, impaired clearance of intracellular pathogens, dysregulated IFN-γ secretion, and production of fibrogenic chemokines [148].

Disruption of the gut–liver axis—exacerbated by dysbiosis and microbial translocation—further amplifies the inflammatory milieu and contributes to the loss of immune tolerance [10,190]. Although these processes are also observed in HIV-negative individuals with metabolic syndrome, they are accentuated in PLWH due to prior immune depletion, viral latency, and residual chronic inflammation, forming a clinical model of prematurely and qualitatively divergent immune aging [9,191].

In summary, metabolic dysfunction in PLWH is not merely a parallel comorbidity but a co-driver of immunosenescence. Integrating metabolic, hepatic, and immunologic biomarkers is essential for early identification of high-risk individuals and for developing targeted interventions to restore immune resilience.

Figure 7 schematically integrates the main pathogenic pathways linking MASLD, T2DM, and HIV to immune aging, emphasizing the roles of ROS, IL-6, TNF-α, and regulatory T-cell depletion in driving inflammaging and CD8^+^ T-cell dysfunction.

### 5.4. Comparison with HIV-Negative Populations: Physiological vs. Accelerated Aging

The comparison between physiological immune aging and that observed in PLWH undergoing effective antiretroviral therapy reveals substantial quantitative and qualitative differences. Whereas immunosenescence associated with chronological aging follows a gradual trajectory—marked by thymic involution, reduction in naïve T cells, and accumulation of effector memory subsets—such changes occur prematurely and more intensively in PLWH, even among young individuals with suppressed viremia [192,193].

Cross-sectional and longitudinal studies have shown that many PLWH exhibit an immune profile comparable to that of HIV-negative individuals 20–30 years older: inverted CD4/CD8 ratio, expansion of CD8^+^CD28^−^CD57^+^ T cells, elevated systemic inflammatory markers (IL-6, D-dimer, sCD14), and diminished vaccine responses—particularly to T-cell-dependent antigens such as influenza, HBV, and SARS-CoV-2 [194,195].

Unlike healthy older adults, in whom inflammaging is generally restrained by active regulatory mechanisms, PLWH experience persistent immune activation and microbial translocation, sustaining a state of heightened chronic inflammation. This is accompanied by increased expression of interferon-stimulated genes (ISGs), chemokines such as CXCL9/CXCL10, and inflammasome activation [129].

The result is an accelerated deterioration of immune function and earlier onset of immune-inflammatory comorbidities, including frailty, osteoporosis, cognitive decline, and cardiovascular disease.

Beyond immunophenotypic alterations, epigenetic markers also confirm this discrepancy. In PLWH, molecular clocks (e.g., GrimAge, PhenoAge) are significantly advanced relative to chronological age, with epigenetic aging rates exceeding 5–7 biological years per decade of infection, especially in individuals with concomitant metabolic dysfunction [196].

This differential profile has led to the conceptualization of “accentuated aging” in PLWH—a term that captures not merely a chronological acceleration of typical aging traits but also a qualitative deviation from the physiological trajectory, characterized by atypical immune phenotypes, metabolic exhaustion, and reduced resilience [197].

These findings suggest that HIV-induced aging represents a unique and informative model for studying immunosenescence—potentially translatable to non-infectious, metabolically compromised contexts. Indeed, the systematic comparison between PLWH and HIV-negative individuals undergoing physiological aging provides an ideal platform to test geroscience-based interventions, assess predictive biomarkers, and design multimodal preventive strategies grounded in biological and functional, rather than chronological, indicators.

## 6. Clinical Implications

### 6.1. Increased Infectious Risk: Recurrent and Severe Infections in Older Adults and PLWH

Immunosenescence is a key determinant of infectious vulnerability in both older adults and people living with HIV (PLWH), even when there is an apparent quantitative recovery of T cells. The decline in both innate and adaptive immune responses compromises the body’s ability to recognize, eliminate, and control pathogens, resulting in an increased frequency, severity, and atypical presentation of infections [198].

In older adults, the reduction in naïve T cells, clonal expansion of senescent CD8^+^ T lymphocytes, loss of TCR diversity, and impaired B-cell responses create a dysfunctional immunological profile characterized by low-grade chronic inflammation (“inflammaging”), which hampers effective responses to bacterial and viral infections [73].

Clinically, this translates into a higher incidence of pneumonia, bacteremia, urinary tract infections, and viral reactivations (e.g., herpes zoster, CMV), often associated with worse outcomes and repeated hospitalizations. In PLWH, these mechanisms are both accelerated and intensified. Chronic immune activation, microbial translocation, and intestinal dysbiosis contribute to sustaining a high inflammatory tone, which impairs effector cell function and facilitates the occurrence of opportunistic infections, even in individuals with suppressed viremia and CD4^+^ counts > 500/mm^3^ [172].

Metabolic dysfunction acts as an additional catalyst for this vulnerability. In PLWH with MASLD or type 2 diabetes, lipotoxicity, oxidative stress, and activation of the NLRP3 inflammasome amplify the production of inflammatory cytokines (IL-6, TNF-α), with deleterious effects on mitochondrial function and immune homeostasis [199]. Accumulation of ROS, reduced phagocytic capacity, and regulatory T-cell dysfunction impair pathogen clearance and increase the risk of severe infections, particularly respiratory and systemic. Phenotypic alterations such as expansion of CD8^+^CD28^−^CD57^+^ T cells, diminished IFN-γ production, persistent monocyte activation, and loss of B-cell diversity contribute to an ineffective and poorly coordinated immune response [200].

Moreover, markers of inflammaging such as CXCL9 and IL-1β are elevated in PLWH with altered metabolic profiles, suggesting a heightened predisposition to vascular frailty and functional immunodeficiency [95].

In clinical practice, these mechanisms result in a higher incidence of recurrent infectious events, including bronchopulmonary exacerbations, complicated urinary tract infections, refractory skin infections, and sepsis—often with atypical or prolonged courses—even in patients considered immunologically “reconstituted”. The coexistence of advanced age, metabolic dysfunction, and HIV thus constitutes a high-risk pathogenic triad for infection, requiring personalized monitoring strategies and integrated approaches to preventive medicine [201].

### 6.2. Reduced Vaccine Response: Evidence from Influenza, Pneumococcal, and SARS-CoV-2 Vaccines

The reduction in immunocompetence observed in older adults and PLWH is reflected in decreased vaccine efficacy, both in terms of antibody production and cellular immunity. Immunosenescence, characterized by the loss of lymphocyte clonal diversity, expansion of exhausted T cells, and B-cell dysfunction, undermines the body’s ability to mount an adequate response to vaccine antigens—particularly in settings of chronic inflammation and metabolic dysfunction [202].

In older adults, a marked decline in the response to influenza and pneumococcal vaccines has been documented, with lower antibody titers, shorter duration of protection, and reduced production of high-affinity IgG [173].

This phenomenon is attributable to impaired function of follicular helper T cells, depletion of naïve lymphocytes, and expansion of senescent B cells [203].

In PLWH, the situation is further compounded by persistent immunophenotypic alterations, even under conditions of virologic suppression. The response to influenza, pneumococcal, and HBV vaccines is frequently suboptimal, particularly in individuals with low CD4^+^ nadir, inverted CD4/CD8 ratios, or metabolic comorbidities [204]. With the introduction of SARS-CoV-2 vaccines, particularly mRNA and bivalent formulations, it has become possible to more precisely characterize the immunological profile of vaccine responses in PLWH [205].

Most individuals with CD4^+^ counts > 350/mm^3^ and suppressed viremia initially develop an adequate antibody response, but with lower titers and more rapid decline compared to the general population, especially regarding neutralizing IgG production and specific T-cell memory [206].

In the presence of metabolic dysfunction, the scenario is even more critical. PLWH with MASLD or documented insulin resistance exhibit weaker antibody responses and reduced antigen-specific IFN-γ production. The TyG index has been found to inversely correlate with both the intensity and duration of the anti-SARS-CoV-2 vaccine response, emerging as a potential predictive marker of vaccine hyporesponsiveness [32].

To date, data on bivalent SARS-CoV-2 vaccines in PLWH with metabolic dysfunction remain limited, but preliminary studies suggest that responses are still attenuated compared to HIV-negative controls, with reduced expansion of effector T cells and low-avidity neutralizing antibodies [207,208].

Immunometabolic immunosenescence thus emerges as a critical determinant of reduced vaccine efficacy.

In response to these challenges, alternative vaccination strategies have been proposed for PLWH with a senescent phenotype or high-risk metabolic profile: enhanced dosing, early boosters, use of TLR-targeted adjuvants, and personalized vaccination schedules based on biological markers such as the CD4/CD8 ratio, CXCL10 levels, or the iAge index [95]. However, the clinical efficacy of such strategies remains to be confirmed in large-scale controlled studies.

In summary, the interplay between HIV, metabolic dysfunction, and immunosenescence creates a context of complex vaccine hyporesponsiveness, calling for a redefinition of immunization strategies in vulnerable populations—guided by immunometabolic parameters rather than clinical criteria alone.

### 6.3. Healthy Aging and Immune Resilience: Predictive Models and Preventive Strategies

In a clinical landscape increasingly shaped by population aging and the chronic course of HIV infection, the concept of “healthy aging” acquires strategic relevance. Rather than focusing solely on survival, the aim shifts toward preserving immune competence, metabolic resilience, and functional quality of life [209].

In this context, immune resilience can be understood as the immune system’s ability to sustain effective, regulated, and adaptive responses despite aging or the presence of chronic conditions such as HIV, MASLD, or type 2 diabetes [210]. Predictive models based on epigenetic, immunophenotypic, and metabolic markers are gradually replacing chronological age as the principal parameter for assessing biological risk. Among these, epigenetic clocks (Horvath, Hannum, GrimAge, PhenoAge) provide accurate estimates of biological age, while the iAge index—developed using deep learning techniques applied to inflammatory panels—has demonstrated significant associations with frailty, multimorbidity, and functional decline, both in PLWH and individuals with metabolic dysfunction [95].

Other functional biomarkers, such as the CD4/CD8 ratio, expansion of CD28^−^CD57^+^ T cells, circulating levels of CXCL9/CXCL10, and the TyG index, offer critical insights into immunometabolic vulnerability, enabling early risk stratification and the identification of subclinical senescent phenotypes [32].

Non-pharmacological interventions have proven effective in slowing the progression of immunosenescence. Regular physical activity enhances NK-cell function, TCR diversity, and mitochondrial biogenesis, while moderate caloric restriction is associated with systemic reductions in IL-6 and TNF-α levels, as well as a deceleration of epigenetic clocks [211]. Targeted nutrition—or immunonutrition—represents another promising pillar. Supplementation with polyphenols, omega-3 fatty acids, soluble fibers, and antioxidant micronutrients has shown beneficial effects on the metabolic plasticity of regulatory T cells and on modulation of the SASP phenotype [212].

In parallel, prebiotic and probiotic supplementation aimed at restoring gut eubiosis has demonstrated potential benefits in reducing microbial translocation and attenuating systemic inflammation in PLWH [172,183].

Metformin, in addition to improving insulin resistance, has shown significant immunomodulatory properties. It modulates mTOR/AMPK signaling in T cells, inhibiting chronic activation and the production of pro-inflammatory cytokines such as IFN-γ, IL-2, and TNF-α [213].

In CD8^+^ cells, it promotes a shift toward memory phenotypes (CD8^+^CXCR3^+^ T_CM) and protects against cellular senescence by enhancing telomerase activity and regenerative capacity. Moreover, in HIV-positive adults with immune dysfunction, metformin supports reduced expression of exhaustion markers in CD4^+^ T cells and promotes immune reconstitution [214,215].

GLP-1 receptor agonists (liraglutide, semaglutide), already employed in the treatment of diabetes and MASLD, exhibit pleiotropic effects on hepatic steatosis reduction, microbiota composition, and inflammatory cytokine secretion [216].

The integration of decision-making algorithms based on neural networks—incorporating epigenetic, immunologic, and metabolic data—may, in the future, guide personalized predictive interventions, even in complex clinical settings such as PLWH with metabolic comorbidities. The systematic adoption of these tools within preventive medicine pathways represents both an organizational and cultural challenge but also a tangible opportunity to transform vulnerability into immune resilience.

## 7. Future Perspectives and Research Directions

### Integrated Models for Immunometabolic Risk Assessment

The early identification of immunometabolic risk in PLWH and in HIV-negative individuals with metabolic dysfunction represents one of the central challenges in aging medicine. Traditional approaches based on chronological age and individual comorbidities are increasingly inadequate at capturing the complexity of individual biological trajectories. The emergence of integrated biomarkers—epigenetic, inflammatory, immunophenotypic, and metabolic—has paved the way for the development of multidimensional predictive models capable of stratifying risk with greater precision and timeliness.

One of the most innovative tools in this field is the iAge index, developed by Sayed et al., which combines circulating inflammatory profiles (CXCL9, IL-6, IL-1β, TNF-α) with neural networks trained on longitudinal data to estimate immune age and predict the likelihood of frailty, multimorbidity, and functional decline [95].

In PLWH, iAge is significantly elevated compared to HIV-negative controls, regardless of virologic suppression, and is strongly associated with clinical parameters of cardiovascular risk and cognitive decline. Another composite tool, the IMM-AGE score developed by Alpert et al., integrates transcriptional and phenotypic data to define a functional immune status that is more informative than chronological age or lymphocyte count alone [88]. This index is particularly useful in assessing risk in immunologically “reconstituted” individuals who nonetheless exhibit persistent inflammatory activation. The integration of these markers with established metabolic parameters enables even more accurate stratification. The TyG index (triglyceride–glucose), a surrogate marker of insulin resistance, has shown significant associations with the CD4/CD8 ratio, expansion of senescent T cells, and hepatic steatosis in PLWH, outlining a high-risk immunometabolic phenotype [32].

The use of the TyG index as an accessible and reproducible marker is especially promising in resource-limited clinical settings. Similarly, hepatometabolic parameters such as the CAP score (controlled attenuation parameter), fibrosis index (FIB-4), and the presence of MASLD can be incorporated into predictive models, as they are associated with immunosenescence, inflammaging, and functional decline in PLWH [217].

The application of machine learning algorithms trained on multi-omic (epigenomic, transcriptomic, immunomic) and clinical data enables the identification of early vulnerability patterns, even in apparently compensated individuals. Recent studies suggest that personalized combinations of indicators (e.g., iAge + TyG + CXCL9) may outperform single biomarkers in predicting clinical events [218].

Finally, the digitalization of medical records and the integration of immunometabolic data into real-time predictive platforms enable the development of dynamic clinical scores that can be updated longitudinally, supporting both early intervention for at-risk individuals and monitoring of intervention efficacy [219].

In summary, integrated risk assessment models represent a paradigm shift in the management of aging in HIV and metabolic dysfunction. By shifting the focus from episodic comorbidity management to proactive prevention of immune vulnerability, they lay the foundation for predictive, personalized medicine that transcends traditional disciplinary boundaries.

## 8. Conclusions

The integrated analysis of clinical, molecular, and immunological evidence presented in this review confirms the tight interconnection between metabolic dysfunction, immunosenescence, and HIV infection as converging determinants of accelerated and dysfunctional immune aging. In people living with HIV (PLWH), even under stable virologic suppression, a profoundly altered immune profile is observed, characterized by persistent activation, loss of clonal diversity, expansion of senescent T cells, and decline of adaptive competence. These alterations are exacerbated by the coexistence of MASLD, insulin resistance, and low-grade chronic inflammation, defining a vulnerable immunometabolic phenotype at high risk for infections, vaccine hyporesponsiveness, multimorbidity, and early-onset frailty. The reconceptualization of hepatic steatosis as an epiphenomenon of systemic metabolic and immunological dysfunction has redefined the role of the liver as an integrative organ—rather than a passive target—within the gut–metabolism–immunity axis. Mitochondrial alterations, NLRP3 inflammasome activation, microbial translocation, and dysbiosis contribute to the development of pathological inflammaging, a phenomenon shared by older individuals and PLWH. In this context, biological age—not chronological age—emerges as the new paradigm for risk assessment, patient stratification, and the design of personalized preventive strategies. Integrated biomarkers such as iAge, the TyG index, and IMM-AGE score offer promising tools to detect senescent immune phenotypes early and to guide targeted interventions based on geroscientific models and combined immunometabolic approaches. Among the emerging therapeutic targets, metformin, GLP-1 receptor agonists, senolytic agents, and immunonutrition show potential for modulating immune plasticity, mitochondrial function, and microbiota composition. In light of current evidence, a revision of the clinical paradigm for managing HIV-positive individuals—particularly those with metabolic dysfunction or advanced age—is warranted. The shift should move from a virologically centered approach to one based on predictive and preventive medicine, grounded in integrated biological markers, with the aim of preserving immune resilience and promoting healthy aging.

### Limitations of the Review

This review is narrative in nature and based on a structured but non-systematic selection of literature, focused on seven predefined thematic sources. While ensuring thematic coherence and depth, this approach may exclude relevant data published outside the selected documents. Quantitative meta-analyses and critical evaluations of the methodological quality of cited studies were not included. Additionally, much of the evidence concerning therapeutic interventions (e.g., senolytics, FMT, tirzepatide) derives from preclinical models or pilot trials with limited sample sizes, restricting the generalizability of the findings. Finally, the heterogeneity of the PLWH population and the lack of a standardized definition of immunosenescence present conceptual challenges for the uniform application of the proposed models.

Despite these limitations, the adopted approach allows for a clear delineation of shared pathophysiological trajectories linking HIV, metabolic dysfunction, and immune aging and suggests innovative clinical and research perspectives. Future longitudinal, multidimensional studies are needed to validate the predictive biomarkers discussed and to assess the effectiveness of proposed strategies in well-characterized cohorts stratified by biological risk.

## Figures and Tables

**Figure 1 biomedicines-13-02513-f001:**
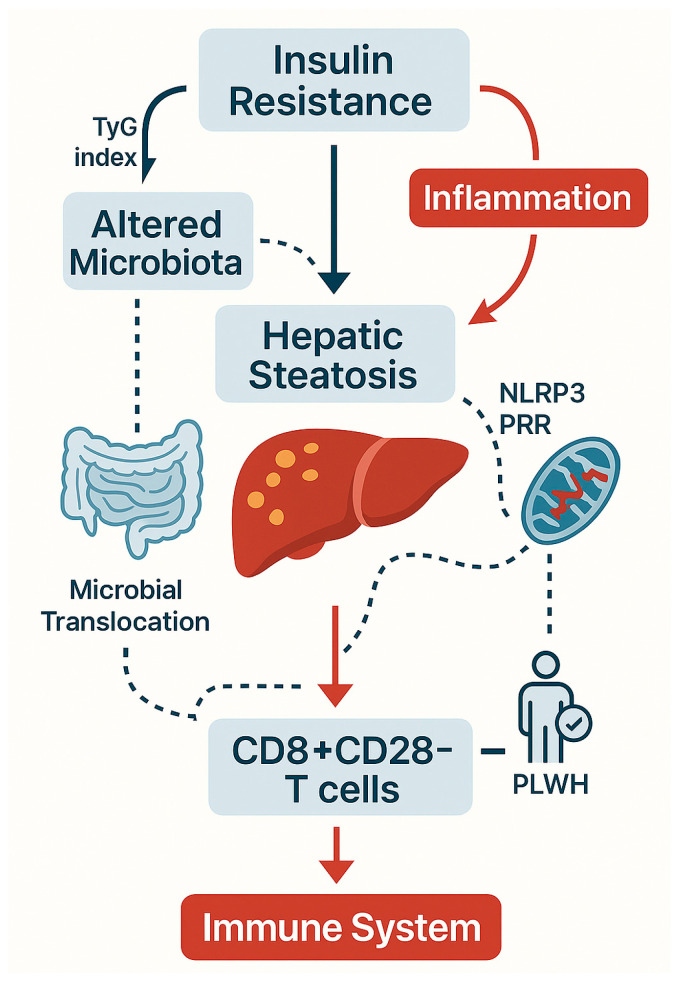
Schematic representation of the immunometabolic cascade linking insulin resistance, gut dysbiosis, and hepatic steatosis to immune senescence in people living with HIV (PLWH). Insulin resistance promotes hepatic steatosis and inflammation, both directly and through microbiota alterations. Dysbiosis and microbial translocation further activate hepatic pattern recognition receptors (PRRs) and the NLRP3 inflammasome, driving mitochondrial dysfunction and CD8^+^CD28^−^ T-cell accumulation. This process culminates in immune dysregulation and accelerated immunosenescence in PLWH. The TyG index is shown as a surrogate marker of insulin resistance associated with this pathway. Solid arrows indicate direct causal pathways; dashed lines indicate indirect or modulatory interactions. Image created using GraphPad Prism (v10.2.2), and Microsoft PowerPoint (Office 365, Build 2405).

**Figure 2 biomedicines-13-02513-f002:**
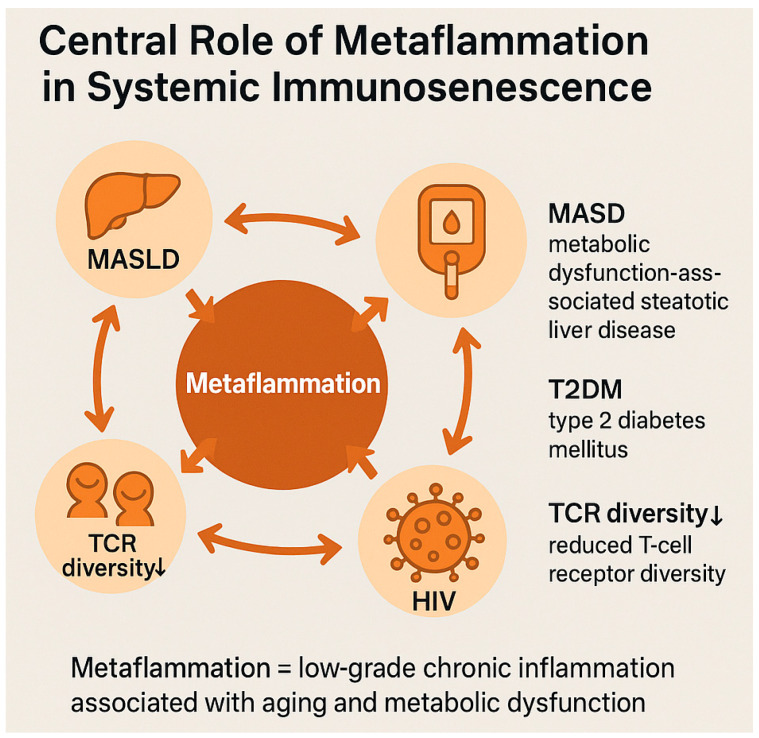
Schematic representation of the immunosenescence cascade driven by metaflammation. Metaflammation—chronic low-grade inflammation that is associated with metabolic dysfunction—serves as a central pathogenic axis linking various immunometabolic conditions. The diagram illustrates reciprocal interactions between metaflammation and clinical states including metabolic dysfunction-associated steatotic liver disease (MASLD), type 2 diabetes mellitus (T2DM), human immunodeficiency virus (HIV) infection, and exposure to damage-associated molecular patterns (DAMPs). Each of these conditions contributes to, and is exacerbated by, metaflammation through shared mediators such as interleukin-6 (IL-6), soluble CD14 (sCD14), and reactive oxygen species (ROS). Reduced T-cell receptor (TCR) diversity is shown as a hallmark of immune aging, further promoting the cycle of chronic inflammation and immune exhaustion. Arrows indicate bidirectional, self-reinforcing interactions between metaflammation and the associated conditions. This model emphasizes the centrality of metaflammation in the pathogenesis of systemic immunosenescence. Image created using GraphPad Prism (v10.2.2) and Microsoft PowerPoint (Office 365, Build 2405).

**Figure 3 biomedicines-13-02513-f003:**
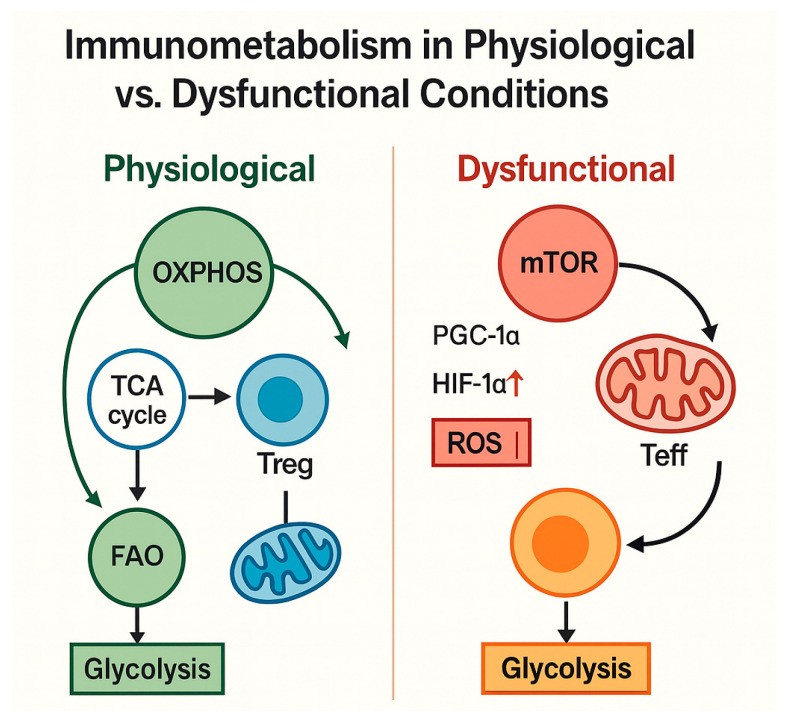
Comparison of immunometabolic pathways in physiological versus dysfunctional conditions. Under physiological homeostasis, immune cells rely on oxidative phosphorylation (OXPHOS), the tricarboxylic acid (TCA) cycle, and fatty acid oxidation (FAO) to support regulatory T-cell (Treg) function and maintain metabolic plasticity. In contrast, dysfunctional states—characterized by activation of mTOR signaling, reduced PGC-1α expression, and upregulation of HIF-1α—promote a shift toward glycolysis and increased reactive oxygen species (ROS) production, driving effector T-cell (Teff) hyperactivation, and immunometabolic exhaustion. This dichotomy underlies the progression from immune balance to senescence. Arrows indicate the direction of metabolic fluxes and functional interactions between pathways and immune cell phenotypes. Image created using GraphPad Prism (v10.2.2) and Microsoft PowerPoint (Office 365, Build 2405).

**Figure 4 biomedicines-13-02513-f004:**
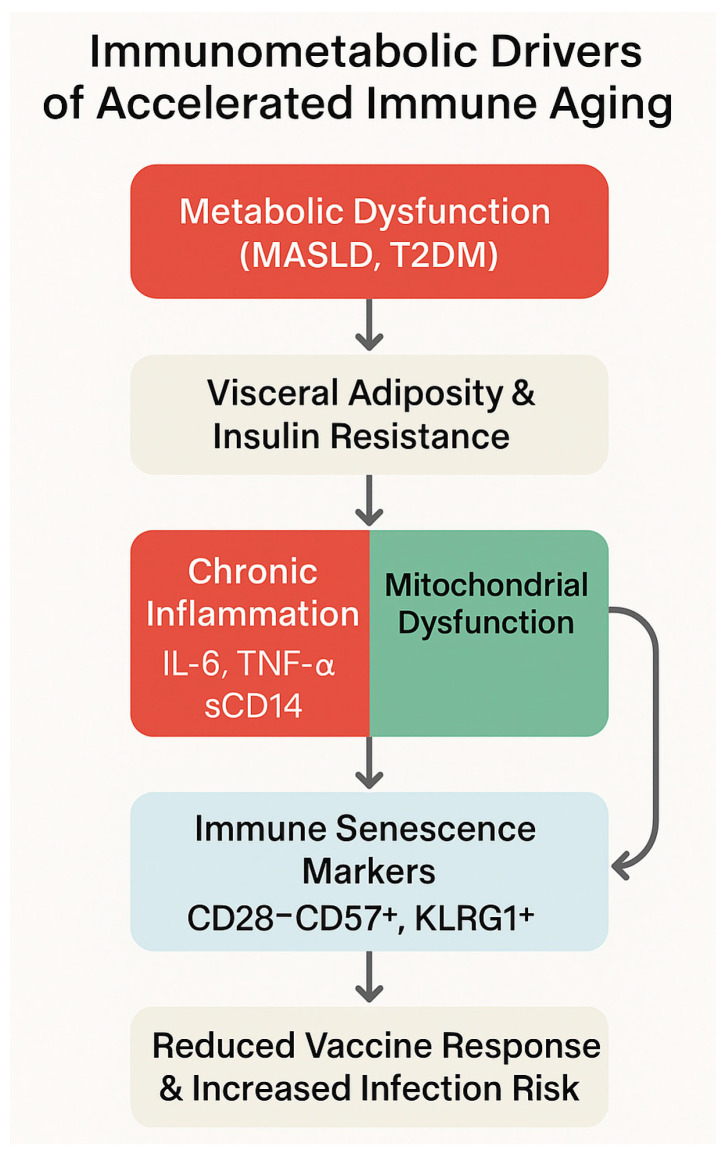
Immunometabolic mechanisms contributing to accelerated immune aging. The figure illustrates a sequential pathway beginning with metabolic dysfunction, including metabolic dysfunction-associated steatotic liver disease (MASLD) and type 2 diabetes mellitus (T2DM), leading to visceral adiposity and insulin resistance. These metabolic alterations promote chronic inflammation—mediated by IL-6, TNF-α, and sCD14—and mitochondrial dysfunction, which synergistically drive immune senescence. Senescent immune phenotypes are marked by CD28^−^CD57^+^ and KLRG1^+^ expression. The culmination of these processes results in diminished vaccine responsiveness and heightened susceptibility to infections. Image created using GraphPad Prism (v10.2.2) and Microsoft PowerPoint (Office 365, Build 2405).

**Figure 5 biomedicines-13-02513-f005:**
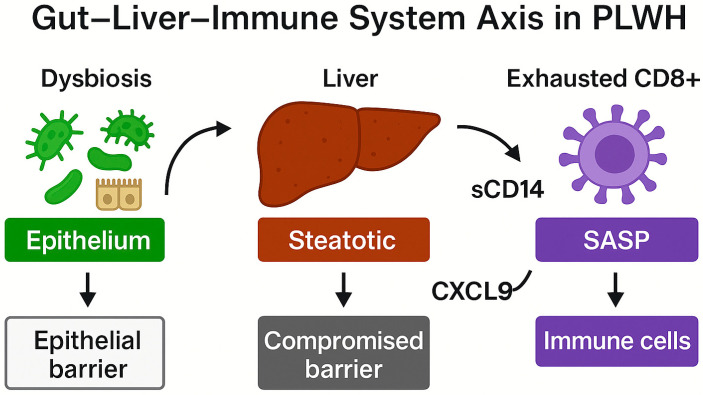
Schematic representation of the gut–liver–immune axis disruption in people living with HIV (PLWH). Intestinal dysbiosis and epithelial barrier dysfunction facilitate microbial translocation, allowing lipopolysaccharide (LPS) and other pathogen-associated molecules to reach the liver. In the presence of hepatic steatosis, the liver becomes a pro-inflammatory hub, compromising epithelial integrity and promoting the release of soluble CD14 (sCD14) and CXCL9. These inflammatory mediators contribute to CD8^+^ T-cell exhaustion and the development of a senescence-associated secretory phenotype (SASP), which sustains systemic immune activation and functional decline. This diagram highlights the bidirectional interactions among dysbiosis, liver inflammation, and immune senescence in PLWH. Image created using GraphPad Prism (v10.2.2) and Microsoft PowerPoint (Office 365, Build 2405).

**Figure 6 biomedicines-13-02513-f006:**
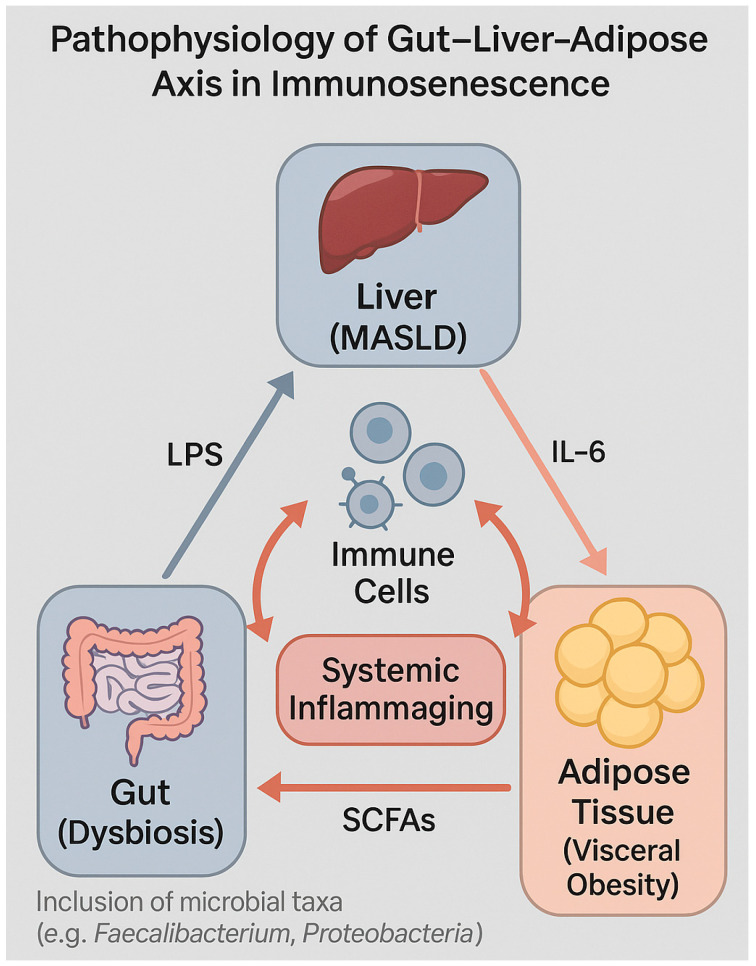
Pathophysiological interactions within the gut–liver–adipose axis contributing to systemic inflammaging and immune senescence. The diagram illustrates how dysbiosis in the gut promotes translocation of lipopolysaccharide (LPS), which activates immune cells and contributes to hepatic inflammation in metabolic dysfunction-associated steatotic liver disease (MASLD). Simultaneously, visceral adipose tissue secretes IL-6, reinforcing the pro-inflammatory environment. These stimuli collectively drive systemic inflammaging and immune cell dysregulation, with feedback effects that perpetuate microbial imbalance and immunometabolic dysfunction. Image created using GraphPad Prism (v10.2.2) and Microsoft PowerPoint (Office 365, Build 2405).

**Figure 7 biomedicines-13-02513-f007:**
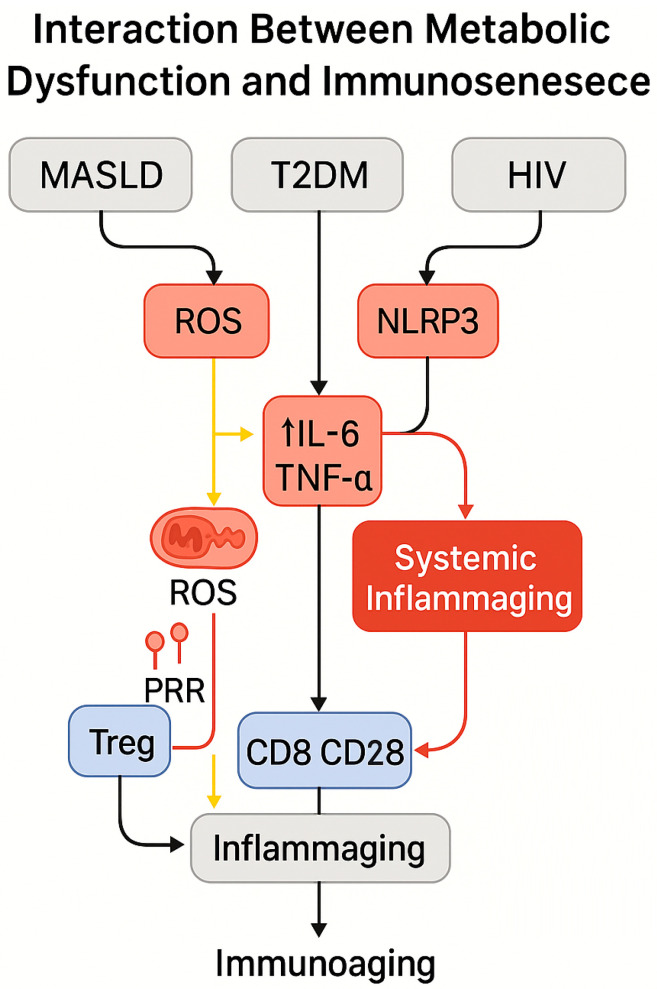
Mechanistic model illustrating the convergence of metabolic dysfunction and HIV infection in promoting immune aging. Metabolic dysfunction-associated steatotic liver disease (MASLD), type 2 diabetes mellitus (T2DM), and HIV infection each contribute to increased production of reactive oxygen species (ROS), activation of the NLRP3 inflammasome, and secretion of pro-inflammatory cytokines such as IL-6 and TNF-α. These events drive systemic inflammation, impair regulatory T-cell (Treg) function, and activate pattern recognition receptors (PRRs), culminating in CD8^+^ T-cell senescence (CD28^−^), reduced CD2 signaling, and expansion of the inflammaging phenotype. The resulting immune remodeling contributes to accelerated immune aging. Arrows indicate causal pathways and functional interactions within the model. Image created using GraphPad Prism (v10.2.2) and Microsoft PowerPoint (Office 365, Build 2405).

**Table 1 biomedicines-13-02513-t001:** Early Drivers of Immunosenescence: HIV and Metabolic Dysfunction.

Pathogenic Driver	Description	Observations in PLWH	References
Metabolic dysfunction	Includes insulin resistance, visceral obesity, MASLD, T2DM	Highly prevalent despite viral suppression	[2,3,4]
Visceral adipose tissue	Pro-inflammatory secretome (IL-6, TNF-α, leptin); ROS and DAMPs	Induces senescent immune activation	[5]
HIV-related immune remodeling	CD4/CD8 inversion, CD8^+^CD28^−^ accumulation, naive T-cell loss	Persistent despite effective ART	[6,7]
Gut–liver–immune axis	Microbial translocation, chronic hepatic inflammation	Altered mucosal barrier, liver dysfunction	[9,10]
Immune aging biomarkers	Epigenetic clocks, IMM-AGE, TyG index	Predict frailty and cardiometabolic risk	[11,12]

PLWH: people living with HIV; MASLD: metabolic dysfunction-associated steatotic liver disease; T2DM: type 2 diabetes mellitus; ROS: reactive oxygen species; DAMPs: damage-associated molecular patterns; IMM-AGE: immune age score; TyG index: triglyceride–glucose index; ART: antiretroviral therapy.

**Table 2 biomedicines-13-02513-t002:** MASLD, T2DM, and Chronic Low-Grade Inflammation.

Condition	Pathophysiological Features	Immune Consequences in PLWH	References
MASLD	FFA overload, lipotoxicity, ROS, NLRP3 activation	CD8^+^ and NKT senescence; CD4/CD8 ratio decline	[13,14,15,16,17,18,24,25]
Mitochondrial dysfunction	Impaired biogenesis (PGC-1α, TFAM), increased oxidative stress	Reduced immune metabolic flexibility	[26,27]
T2DM	Chronic hyperglycemia, adipokine imbalance, systemic metaflammation	Expansion of CD28^−^/CD57^+^ T cells, naive T-cell loss	[30,31,32,33,34,35,36]
Metaflammation mediators	DAMPs, mtDNA, ATP, ROS activating PRRs and inflammasomes	Amplifies systemic senescence in PLWH	[37,38,39,40]
TyG index	Surrogate for insulin resistance; correlates with senescent T cells	Proposed immunometabolic risk biomarker	[29,43,44]

PLWH = people living with HIV; MASLD = metabolic dysfunction-associated steatotic liver disease; T2DM = type 2 diabetes mellitus; FFA = free fatty acid; ROS = reactive oxygen species; NLRP3 = nucleotide-binding oligomerization domain-like receptor family pyrin domain containing 3; PGC-1α = peroxisome proliferator-activated receptor gamma coactivator 1-alpha; TFAM = mitochondrial transcription factor A; DAMPs = damage-associated molecular patterns; mtDNA = mitochondrial DNA; ATP = adenosine triphosphate; PRRs = pattern recognition receptors; TyG index = triglyceride–glucose index.

**Table 3 biomedicines-13-02513-t003:** Immunosenescence Mechanisms: Cellular and Molecular Basis.

Immune Component	Senescent Features	HIV-Related Amplification	References
Innate immunity	Reduced neutrophil chemotaxis, impaired NK cytotoxicity	Chronic activation, low IFN-γ output	[73,74,75]
Adaptive T cells	CD8^+^CD28^−^CD57^+^KLRG1^+^ expansion, TCR loss, apoptosis resistance	Persistent even with ART	[76,77,78,79,80]
B cells	Decreased naive pool, increase in ABCs, poor antibody response	Impaired vaccination efficacy	[78,79]
Molecular markers	Epigenetic clocks, telomere shortening, IMM-AGE, iAge, TyG	Accelerated epigenetic aging in PLWH	[87,91,92,93,94]
Functional consequences	Reduced immune coordination (“integrated immunocompetence”)	Elevated frailty, multimorbidity	[83,85]

PLWH: people living with HIV; NK: natural killer; TCR: T-cell receptor; ABCs: age-associated B cells; ART: antiretroviral therapy; IMM-AGE: immune age metric based on immune phenotyping; TyG: triglyceride–glucose index.

**Table 4 biomedicines-13-02513-t004:** Immunometabolism and Cellular Bioenergetics in MASLD and T2DM.

Metabolic Disruption	Immune Impact	Observations in PLWH with MASLD/T2DM	References
Loss of immune plasticity	Impaired glycolysis–OXPHOS switch; ATP deficit	Promotes senescent, exhausted phenotypes	[120,121,122,123]
Steatotic liver environment	ROS, IL-1β/IL-6 secretion, NLRP3 activation	Dysfunctional CD8^+^/NKT cells with PD-1^+^/CD57^+^	[130,131,132]
Mitochondrial dysfunction	Reduced PGC-1α, TFAM, mitophagy dysregulation	Metabolic exhaustion of immune cells	[27,131]
T2DM-induced reprogramming	Persistent mTOR/HIF-1α activation; glycolytic overload	SASP phenotypes, Treg depletion	[133,134]
TyG index	Captures interplay of hepatic steatosis and insulin resistance	Tracks CD4/CD8 inversion and senescent T cells	[29]

PLWH: people living with HIV; NKT: natural killer T cells; Treg: regulatory T cells.

**Table 5 biomedicines-13-02513-t005:** Gut–Liver Axis and Systemic Inflammation in HIV+ Individuals.

Axis Component	Dysregulation Mechanism	Immunological Repercussions in PLWH	References
Adipose tissue	Leptin/resistin ↑, adiponectin ↓; M1 macrophage recruitment	Chronic metaflammation and cytokine storm	[135,136,137]
Dysbiosis	SCFAs depletion, pro-inflammatory taxa enrichment	Loss of mucosal tolerance, immune senescence	[139,140]
Microbial translocation	LPS and PAMPs reach liver, activate PRRs	Exacerbates hepatic and systemic inflammation	[140,141]
SCFA modulation	Butyrate activates GPR41/43 on immune cells	T-cell memory and regulatory circuits affected	[142]
ART effects	Induces lipodystrophy, worsens metabolic and microbiota balance	Accelerates immunometabolic decline	[138]

PLWH: people living with HIV; PRRs: pattern recognition receptors; SCFAs: short-chain fatty acids; LPS: lipopolysaccharide; PAMPs: pathogen-associated molecular patterns; ART: antiretroviral therapy. ↑ indicates increase; ↓ indicates decrease.

**Table 6 biomedicines-13-02513-t006:** Chronic Immune Activation and Accelerated Aging in HIV.

Mechanism	Immunological Consequences	Clinical Relevance	References
Persistent immune activation	CD4^+^ depletion, CD8^+^ hyperactivation (CD28^−^CD57^+^), IFN-γ ↑, TNF-α ↑	Senescence phenotype despite ART	[145,146,147]
B- and NK-cell dysfunction	Loss of memory B cells, ABCs ↑; NK receptor downregulation	Impaired surveillance and vaccine response	[148,149]
Dendritic cell impairment	Decreased numbers, poor antigen presentation	Deficient priming of naive T cells	[151]
CMV co-infection	Clonal expansion of senescent CD8^+^ cells	Immune exhaustion and frailty	[152]
Microbial translocation	Sustains inflammation (↑ sCD14, LBP, I-FABP)	Predictor of mortality and comorbidity	[155]

NK: natural killer; ABCs: age-associated B cells; CMV: cytomegalovirus; sCD14: soluble CD14; LBP: lipopolysaccharide-binding protein; I-FABP: intestinal fatty acid-binding protein; ART: antiretroviral therapy. ↑ indicates increase.

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
