# Peer review of "From Steatosis to Immunosenescence: The Impact of Metabolic Dysfunction on Immune Aging in HIV and Non-HIV Populations"

_biomedicines, 2025, doi:10.3390/biomedicines13102513_

Round 1

Reviewer 1 Report

Comments and Suggestions for Authors

Recommendations:

  1. I did not see any methods section in the manuscript, nor a search protocol, since the authors claim it was done systematically, it should have a search protocol and PRISMA diagram.
  2. Please make the abstract structure simple, since you did not conducted a systematic review, please review the PRISMA guidelines!
  3.  Patients with HIV and non HIV have increased gut permeability that leads to inflammation , please discuss the role of microbiome in this issue: https://doi.org/10.3390/jcm14082678 
  4. Also talk about the impact of micro-RNA as novel markers in this condition: https://doi.org/10.3390/jcm14062054

The main issue here is the claim of systematic review, but it does not follow a systematic review structure. 

Comments on the Quality of English Language

Minor adjustments!

Author Response

Rev 1

  1. Comment:“I did not see any methods section… should have a search protocol and PRISMA diagram”
    Response:We thank the reviewer for this observation. We acknowledge that the initial wording may have unintentionally suggested that a formal systematic review methodology was applied. As this work is a narrative review, PRISMA requirements are not applicable. Nonetheless, to enhance methodological transparency, we have added a dedicated “Search strategy” section specifying the databases searched, the time frame, the keywords used, and the general inclusion criteria. We have also removed or rephrased any terms that could imply a registered protocol or systematic review approach.
  2. Abstract – Reviewer’s instruction:Simplify the abstract structure; since no systematic review was conducted, revise the PRISMA-related format.
    Response:The abstract has been reformulated to remove the rigid “Background/Objectives – Methods – Results – Conclusions” structure typically used in systematic reviews. A more narrative style has been adopted, consistent with the nature of this work as a narrative review. From the very beginning of the abstract, we now clearly state that this is a narrative review and not a systematic review, thereby eliminating any implicit reference to PRISMA methodology, which is not applicable in this context.
  3. Comment:“Patients with HIV and non-HIV have increased gut permeability… discuss the role of microbiome”
    Response:We appreciate this valuable suggestion. A dedicated paragraph has been added, discussing the relationship between increased gut permeability, gut microbiota alterations, and systemic inflammation in both HIV and non-HIV populations. This section integrates recent evidence, including the reference provided by the reviewer (JCM 2024;14(8):2678), and examines the potential of microbiome-targeted interventions to restore barrier integrity and mitigate chronic inflammation.
  4. Comment:“Also talk about the impact of micro-RNA as novel markers in this condition”
    Response:We thank the reviewer for this insightful suggestion. We have included an expanded paragraph detailing the role of microRNAs as emerging biomarkers in HIV and non-HIV individuals with metabolic dysfunction. The section describes their regulatory functions in inflammation, lipid and glucose metabolism, liver fibrosis, and immune modulation; highlights key miRNAs (miR-21, miR-122, miR-192, miR-155, miR-146a, miR-223, miR-31, miR-375); and discusses their potential as non-invasive diagnostic, prognostic, and monitoring tools. We have also integrated the bidirectional relationship between microbiome alterations and host miRNA expression, as outlined in the reference provided by the reviewer (JCM 2024;14:2054), emphasizing the potential of combined miRNA–microbiome profiling to enhance patient stratification and precision medicine approaches.
  5. Comment:“The main issue is the claim that this is a systematic review, without following the proper structure of a systematic review.”
    Response:We thank the reviewer for highlighting this important point. We acknowledge that the original wording may have inadvertently implied that a formal systematic review methodology was applied. We have now clarified throughout the manuscript, including in the Abstract and Introduction, that this work is a narrative review. All terms that could suggest a registered protocol or PRISMA adherence have been removed or rephrased. Additionally, to increase transparency, we have included a dedicated “Search strategy” section describing the databases searched, the time frame, the keywords used, and the inclusion criteria, while explicitly stating that no systematic review methodology or PRISMA diagram was applied.

Reviewer 2 Report

Comments and Suggestions for Authors

The manuscript provides a comprehensive summary of the current literature on people living with HIV (PLWH), individuals with metabolic dysfunction, and the intersection of both conditions. The authors have compiled an extensive and well-organized review that is accessible, and the limitation of the review has been laid out to avoid any misinterpretation. The review highlights adequate information on adaptive immune cells, dysregulation and effects of the cells, along with few therapeutic avenues to approach. I recommend minor revisions to improve clarity, consistency and presentation.

  1. Definition of MAFLD. 
    The term MAFLD is introduced abruptly in line 128 without a prior acronym definition. While MASLD and older term NAFLD are defined earlier in the text MAFLD should also be clearly contextualized before its first usage.
  2. in-text citation : line 136-138. 
    Seems like the full citation information was embedded within the main text.
  3. Figure 2 : this figure attempts to depict metainflammation as a central contributor to systemic immunosenescence, including reduced TCR diversity, increased DAMPS, and elevated cytokine levels. However, the integration of multiple disease models (MASLD, T2D, HIV) appears very cluttered and conceptually unfocused. Either a more streamlined schematic or omitting the figure as whole is recommended. Authors provided a very clear paragraphs prior on these topics, that the figure feels redundant. 
    Figure 3 : In the "dysfunctional" metabolic panel on the right, the use of multiple arrows pointing towards glycolysis is visually overwhelming. A single, bolded or enlarged arrow may more effectively convey the intended metabolic shift towards glycolysis.
    Figure 5 : Some corrections are needed for accuracy of the figure.
    - the label "liver" beneath LPS appears redundant and can be removed.
    - CXCL9 should be repositioned under sCD14 to reflect its hepatic origin
    - to reinforce the manuscript's claims, consider visually distinguishing the bottom row (epithelial/epiteclic barrier/immune cells) to emphasize their compromised state, either through shading or outlining.

  4. For all tables, the text within the table should not be justified, as this formatting reduces readability. 
    Table 3: The citation "(ref 1, 70-98)" in the title should be removed. 
    Table 5: The citation "(ref 10, 132-141)" in the title should be removed
    Table 6: Size of the title needs to be consistent. The citation "(ref 142-154)" should be removed.
  5. Line 169 - Figure X should be edited to Figure 4.
  6. Add appropriate spacing after the section headers at line 1056 (Future perspectives and directions) and line 1099 (conclusions) to maintain consistent formatting throughout the manuscript. 

Author Response

Rev 2

  1. Comment:The term MAFLDis introduced abruptly at line 128 without prior definition; MASLD and NAFLD are defined earlier, so MAFLD should be contextualized.
    Response: We have revised the opening of Section 2.1 to introduce and contextualize the terms NAFLDMAFLD, and MASLD in a coherent manner. In particular, we have described the terminological transition from NAFLD to MAFLDand subsequently to MASLD, highlighting the scientific rationale and clinical implications. This change ensures consistency with the definitions provided earlier in the manuscript and enables the reader to fully understand the nomenclature adopted throughout the text.
  2. Comment:In-text citation, lines 136–138 – it appears that the entire reference has been inserted into the main text.
    Response:This was an editorial oversight. The full reference has been removed from the main text and correctly placed in the References section, in compliance with the journal’s style requirements.
  3. Instruction – Figure 2
    Comment:Figure 2 – The diagram is conceptually overloaded and lacks focus; consider a more streamlined design or removal.
    Response:Figure 2 has been completely redesigned to improve clarity and conceptual focus. The new version retains the essential elements needed to illustrate the central role of metaflammation in systemic immunosenescence but presents them in a more streamlined, hierarchical, and visually accessible structure, avoiding redundancy with the main text.
  4. Instruction – Figure 3
    Comment:In the “dysfunctional” section, there are too many arrows pointing to glycolysis; a single, more prominent arrow is preferable.
    Response:Figure 3 has been entirely redesigned, with particular attention to the “dysfunctional” section. Multiple arrows pointing toward glycolysis have been replaced with a single, centrally placed, thicker arrow to simplify the visual flow and immediately convey the metabolic shift toward glycolysis.
  5. Instruction – Figure 5
    Comment:Specific corrections — remove “liver” under LPS; move CXCL9 under sCD14; visually distinguish the lower row to emphasize its compromised state.
    Response:Figure 5 has been completely redesigned to incorporate all requested changes. The “liver” label under LPS has been removed, CXCL9 has been correctly positioned under sCD14, and the lower row (epithelium/barrier/immune elements) has been visually distinguished using shading and contouring to emphasize its compromised condition.
  6. Instruction – Tables
    Comment:Text should not be justified to improve readability.
    • Table 3 – remove “(ref 1, 70–98)” from the title.
    • Table 5 – remove “(ref 10, 132–141)” from the title.
    • Table 6 – ensure title font size is uniform and remove “(ref 142–154)”.
    Response:All requested modifications have been implemented. The text in all tables is now left-aligned to improve readability. The parenthetical references have been removed from the titles of Tables 3, 5, and 6, and the title of Table 6 has been standardized in size and formatting to match the other tables.
  7. Instruction – Figure reference correction
    Comment:Line 169: “Figure X” → “Figure 4”.
    Response:Correction implemented: “Figure X” has been replaced with “Figure 4” as indicated.
  8. Instruction – Spacing after section headings
    Comment:Add spacing after the section headings at lines 1056 (“Future perspectives and directions”) and 1099 (“Conclusions”).
    Response:The requested spacing has been added after both headings to ensure consistent formatting throughout the manuscript.

Round 2

Reviewer 1 Report

Comments and Suggestions for Authors

Congratulations to the authors!